# Perpetrators of gender-based workplace violence amongst nurses and physicians–A scoping review of the literature

**Basnama Ayaz**[1]*, **Graham Dozois**[2], **Andrea L. Baumann**[1], **Adam Fuseini**[1], **Sioban Nelson**[1]

**1** Lawrence S. Bloomberg Faculty of Nursing, University of Toronto, Toronto, Ontario, Canada, **2** Princess Margaret Cancer Centre, Toronto, Canada

* basnama.ayaz@mail.utoronto.ca

**Data Availability Statement:** All relevant data are included in the manuscript and its supporting information files.

## Abstract

In healthcare settings worldwide, workplace violence (WPV) has been extensively studied. However, significantly less is known about gender-based WPV and the characteristics of perpetrators. We conducted a comprehensive scoping review on Type II (directed by consumers) and Type III (perpetuated by healthcare workers) gender based-WPV among nurses and physicians globally. For the review, we followed the Preferred Reporting Items for Systematic and Meta Analyses extension for Scoping Review (PRISMA-ScR). The protocol for the comprehensive review was registered on the Open Science Framework on January 14, 2022, at https://osf.io/t4pfb/. A systematic search in five health and social science databases yielded 178 relevant studies that indicated types of perpetrators, with only 34 providing descriptive data for perpetrators' gender. Across both types of WPV, men (65.1%) were more frequently responsible for perpetuating WPV compared to women (28.2%) and both genders (6.7%). Type II WPV, demonstrated a higher incidence of violence against women; linked to the gendered roles, stereotypes, and societal expectations that allocate specific responsibilities based on gender. Type III WPV was further categorized into Type III-A (horizontal) and Type III-B (vertical). With Type III WPV, gendered power structures and stereotypes contributed to a permissive environment for violence by men and women that victimized more women. These revelations emphasize the pressing need for gender-sensitive strategies for addressing WPV within the healthcare sector. Policymakers must prioritize the security of healthcare workers, especially women, through reforms and zero-tolerance policies. Promoting gender equality and empowerment within the workforce and leadership is pivotal. Additionally, creating a culture of inclusivity, support, and respect, led by senior leadership, acknowledging WPV as a structural issue and enabling an open dialogue across all levels are essential for combating this pervasive problem.

**Funding:** The authors received no specific funding for this work.

**Competing interests:** The authors have declared that no competing interests exist.

## Introduction

The International Labour Office (ILO), the International Council of Nurses (ICN), the World Health Organization (WHO), and Public Services International (PSI) defined WPV as "incidents where staff are abused, threatened or assaulted in circumstances related to their work, including commuting to and from work, involving an explicit or implicit challenge to their safety, well-being or health" [1]. The ILO [2] further defined "gender-based violence and harassment means violence and harassment directed at persons because of their sex or gender, or affecting persons of a particular sex or gender disproportionately".

Irrespective of industry, workplace violence (WPV) can cause lasting trauma and injuries and is a serious threat to human and health resources. WPV includes physical and psychological violence, including physical assault, verbal abuse, sexual harassment, and bullying. Gender-based workplace violence (GB-WPV), which is experienced across operational layers of an organization (horizontal) and organizational hierarchy (vertical), reinforces the differential risk for exposure and outcomes of violence for men and women [3]. Despite extensive research on workplace violence in healthcare, GB-WPV, its perpetrators, and its impact on healthcare professionals remains understudied. We presented the sex-segregated prevalence and risk factors for WPV somewhere else [4]. The earlier paper focused on the scope and scale of workplace violence (WPV), risk factors and its impact on men and women. As part of the same scoping review protocol, this paper reports on GB-WPV perpetrators. It specifically focuses on explaining the root causes of violent acts by individuals and the triggers and circumstances to provide gender-sensitive recommendations.

A systematic review [5] of the consequences of exposure to WPV in the healthcare setting based on 68 studies reported psychological and emotional effects such as post-traumatic stress, depression, anger, and fear. These effects impact work productivity, leading to increased sick leaves, poor job satisfaction, burnout, and higher attrition rates, particularly for women [5]. Studies have also shown that men are more likely to commit physical violence [6] and sexual harassment [7, 8], while women are more often engaged in verbal abuse [9]. Moreover, gender stereotypes and inequalities in the distribution of roles and responsibilities can worsen power imbalances [3]. By recognizing and understanding these issues, employers and organizations can more effectively prevent and deal with gender-based workplace violence, ensuring a safer and more equitable work environment for everyone.

The classification of workplace violence has evolved, delineating distinct categories shaped by the nature of its perpetrators. The current working taxonomy categorizes WPV into four types based on the perpetrators of violence. This typology, as shown in Table 1, emerged from a collaborative effort of a workshop on workplace violence intervention research held in

**Table 1. Type of violence based on perpetrators.**

| | |
|---|---|
| **Type I** | Workplace violence committed by a person who has no legitimate business at the work site" with criminal intent [10]. |
| **Type II** | Workplace violence directed at employees by customers/service consumers, including clients, patients, students, inmates, or visitors, who are patients' companions [10]. |
| **Type III** | Workplace violence against an employee by a present or former employee, supervisor, or manager [10]. |
| | **Horizontal or lateral violence** is "violence, harassment or bullying directed at colleagues who are at equal level within an organization" [11]. |
| | **Vertical violence** perpetrated by senior colleagues, supervisors, and administrative personnel at a higher level than the victim in the organizational or professional hierarchy [11, 12] |
| **Type IV** | Workplace violence is committed by someone who is not an employee but has or is known to have had a personal relationship with an employee [10]. |

Washington, DC, in 2000. The findings of this endeavour were subsequently published by the U.S. Department of Justice in 2001 [13]. Since then, this framework has been adopted by multiple organizations [10, 11, 14], and by researchers [12, 15].

This paper explores the dynamics of workplace violence by categorizing and summarizing both Type II WPV, from patients and significant others, and Type III WPV (horizontal and vertical), which pertain to violence perpetuated by colleagues, supervisors, and administrators within the organization. Additionally, we explore the nuances of GB-WPV, considering both the perpetrators and nurses and physicians as victims of WPV. We summarized perpetrators based on their gender and synthesized the factors attributed to Type II and III, which are prevalent forms of violence reported in the literature. Type I and Type IV violence are beyond the scope of this paper as they focus on a security-based rather than workplace culture interventions. Understanding the factors contributing to these types of WPV is crucial to developing effective preventive interventions and strategies. Currently, there is a dearth of information identifying the characteristics of individuals who are more likely to commit GB-WPV and the characteristics of those targeted by such offences. This review addresses this gap by synthesizing data from studies that reported on the gender/sex data for various forms (please see S1 Text: Definitions of the Forms of Violence) of WPV and perpetrators of WPV among nurses and physicians.

While WPV affects individuals across the gender spectrum and in different professional groups, women are disproportionately affected. Some studies attributed it to their preponderance in the health workforce, their overrepresentation in lower positions in organizational and professional hierarchies, and societal gender norms in most cultures that subjugate women [9, 16]. Recognizing that workplace violence is fundamentally intertwined with broader societal structures rooted in socioeconomic, cultural, and institutional factors, we underscore the necessity of a systematic approach to address this issue—one that is integrated, participatory, culturally and gender-sensitive, and non-discriminatory [1]. While current interventions aimed at addressing WPV primarily focus on assessing the effectiveness of training interventions to prevent and manage WPV in healthcare settings [17–19], they often lack gender-segregated findings for their effectiveness. Clarifying the existing situation on the gender of victims and perpetrators for specific Type/s of violence would help develop gender-sensitive interventions and policies to more effectively address WPV. This scoping review focuses on understanding GB-WPV and its perpetrators in the global health workforce, including nurses and physicians. Our preliminary search for a scoping review revealed that GB-WPV affects men, women, and non-binary persons. However, most studies included in this review reported gender as binary (men and women); only a few studies included non-binary personnel (for sample-see Table 2, in results section). Therefore, we defined *gender* as a binary for this review and deliberated on it in the discussion section.

Our specific objectives set out for this paper were:

1. Describe the proportions of WPV and related perpetrator/s contributing to Type II (from patients/clients/families) and Type III (worker-to-worker) violence among nurses and physicians in different contexts.

2. Summarize the gendered perpetration of Type II and Type III WPV against men and women in the health workforce.

3. Identify gaps in the state of knowledge to recommend direction for future empirical research studies.

**Table 2. Prevalence of various types/forms of workplace violence by gender and types of perpetuators in different clinical settings and professional categories across world.**

| S. # | Author/s, year | Clinical Setting | Professional Category/ies | Sample (Female; Male) | Country/ies | Prevalence of WPV by Type (Female; Male) | Perpetrators | | |
|---|---|---|---|---|---|---|---|---|---|
| | | | | | | | Type 2 | Type 3A | Type 3B |
| 1. | Aghajanloo et al., 2011 [6] | Iran Medical University | Nursing students | 180 (72%; 28) | Iran | No significant relation between students' sex and the frequency of insult (p = 0.051) | X | | X |
| 2. | Brown et al., 2019 [7] | Gynecology | Physicians- members of an international society | 907 (59%; 40%; unknown 1%) | USA and non-USA | Gender discrimination (90%; 72%) Harassment (72.6; 53) Sexual Harassment (84%; 14%) | | | X |
| 3. | Chang et al., 2020 [8] | 4 Universities | Senior nursing students | 310 (87%; 13%) | Taiwan | Sexual Harassment (23.3%; 18.4%) | X | | X |
| 4. | Newman et al., 2021 [9] | Health sector | Healthcare providers, including medical personnel | 294 Data presented: men and women | Uganda | Sexual coercion started during recruitment of health workers and continued after hiring, perpetrated by men in decision-making positions. | | | X |
| 5. | Al-Ghabeesh & Qattom, 2019 [15] | Emergency department | Nurses | 120 (35%; 65%) | Jordan | No significant differences based on the gender of the participant (p = 0.07). | | | X |
| 6. | Sellers et al., 2012 [20] | 19 Healthcare organizations | Registered nurses | 2659 (93%; 7%) | New York State, USA | Women reported significantly greater (p < .05) knowledge of and being a victim of HV than men. | | X | |
| 7. | Alameddin et al., 2015 [21] | Database of the Order of Nurses in Lebanon | Nurses | 593 (79%; 21) | Lebanon | Male nurses had 2.22 times the odds of exposure to physical violence compared to females (95% CI 1.14–4.35, p- 0.019). | X | X | X |
| 8. | Al.Surimi et al., 2020 [22] | 4 hospitals in various regions | Nurses, Physicians, and others | 1075 (86%; 14%) | Saudi Arabia | Bullying (66%; 49%) | X | X | X |
| 9. | Anand et al., 2016 [23] | Tertiary care hospital | Medical residents- various departments | 169 (38.5%; 61.5%) | India | WPV (40%; 41%); Threats (58%; 47%) Physical (4%; 16%); Verbal (17%; 81%) | X | X | X |
| 10. | Boafo et al., 2016 [24] | Hospitals at all levels | Nurses | 592 (79%; 21%) | Ghana | Verbal Abuse (83;17.0%). | X | X | X |
| 11. | Brooks et al., 2022 [25] | Orthopedics | Black orthopedic residents and fellows | 310 Residents (18%; 82%); Attending (22%; 78%) | USA | Micro assaults (65%; 60%) | X | X | X |
| 12. | Campbell et al., 2011 [26] | 4 health care institutions | Nursing personnel (all categories of nurses) | 2166 (91.5%; 8.5%) | USA | Males were nearly twice as likely to have experienced physical WPV compared to females. | X | X | X |
| 13. | Cavalcanti et al., 2018 [27] | PHC | Nurses | 112 (95%; 5%) | Brazil | Workplace violence (72%; 83%) | X | X | X |
| 14. | Ceppa et al., 2020 [28] | Cardiothoracic surgery | Attending surgeons and trainee surgeons | 790 (23%; 75%; Others 2%). | Globally-professional platforms | Sexual Harassment (81%; 46% attending surgeons) Sexual harassment (90%; 32% among trainees) | X | X | X |
| 15. | Chatziionnidis et al., 2018 [29] | 20 NICUs | Nurses and Physicians | 398 (87%; 13%) | Greece | Bullying (56%; 36%) | X | X | X |

*(Continued)*

**Table 2.** (Continued)

| S. # | Author/s, year | Clinical Setting | Professional Category/ies | Sample (Female; Male) | Country/ies | Prevalence of WPV by Type (Female; Male) | Perpetrators | | |
|---|---|---|---|---|---|---|---|---|---|
| | | | | | | | Type 2 | Type 3A | Type 3B |
| 16. | Chen et al., 2021 [30] | Plastic surgery | Plastic surgery trainees | 236 (42%; 54%; no specification 4%) | Pittsburgh- USA | Had been presented provocative imagery/words (42%; 32%). Discomfort from sexually oriented jokes (45%; 33%) | X | X | X |
| 17. | Cheung and Yip, 2017 [31] | Members of the Association of Hong Kong | Nurses | 850 (88%; 12%) | Hong Kong, China | Workplace Violence (44%; 48.6%) | X | X | X |
| 18. | Cho et al., 2020 [32] | Hospital | Early-career hospital nurses | 1171 (86%; 14%) | USA | **Verbal Abuse from patients/family** 1–3 times per month (58%; 49%) Once a week or more (20%; 32.7%) **From Physicians:** 1–3 times per month (38%; 34%) Once a week or more (5%;9%) | | **X** | **X** |
| 19. | Crutcher et al., 2011 [33] | Family medicine | Family medicine residency graduates | 242 (53.2%; 46.4%), Not recorded 0.4%) | Alberta, Canada | IHD by gender (48%; 44%) IHD in the form of work as punishment (20%; 38.6%) IHD in the form of privileges/opportunities being taken away (26.7%; 6.8%) | X | X | X |
| 20. | Dettmer et al., 2021 [34] | Cardiology | Physicians | 567 (49%; 51%) | Germany | Experienced sexual harassment (32%; 7%) | | X | X |
| 21. | Ferri et al., 2020 [35] | Emergency Triage area | Nurses | 27 (44%; 56%) | Italy | Verbal (83%; 100%) Both verbal and physical (17%; 0%) | X | | |
| 22. | Firenze et al., 2020 [36] | Hospitals | Medical personnel | 4545 (57%; 43%) | Italy | Males experience almost three times higher (aOR 2.09, 95% CI 1.51–2.88, p<0.001) | X | X | X |
| 23. | Harthi et al., 2020 [37] | Public hospitals ED | HCWs in ED, including nursing and medical personnel | 324 (66%; 34%) | Saudi Arabia | Workplace violence (42.8%; 57.8%) Physical violence (11%; 20%); Verbal abuse (35%; 47%) | X | X | X |
| 24. | Hu et al., 2019 [38] | General surgery residency programs. | Medical residents | 7409 (39.6%; 59.9%; No data 0.5% | USA | Gender discrimination (65.1%;10.0%). Verbal or emotional Abuse (33.0%; 28.3%) Sexual harassment (19.9%; 3.9%) | X | X | X |
| 25. | Kemper & Schwartz, 2020 [39] | Pediatrics | Pediatric residents- Resident Burnout and Resilience Study Consortium | 1956 (70%; 30) | USA | Any type of Mistreatment 33% (36%; 25%) Bullying19% (20%; 16%) Discrimination 18% (21%; 11%) Sexual Harassment 5.4% (6%; 4%) | X | X | X |
| 26. | Lei et al., 2022 [40] | Emergency department | Nurses | 20136 (89%; 11%) | China | Any type of WPV (89%; 11%) Physical (85%; 15%); non-physical (89%; 11%) | X | X | X |
| 27. | Menhaji et al., 2022 [41] | OBGYN | Trainees | 366 (86.5%; 13.5%) | USA | Sexual harassment (68.7%; 69.6%) | X | X | X |

(Continued)

**Table 2.** (Continued)

| S. # | Author/s, year | Clinical Setting | Professional Category/ies | Sample (Female; Male) | Country/ies | Prevalence of WPV by Type (Female; Male) | Perpetrators | | |
|------|----------------|------------------|---------------------------|-----------------------|-------------|------------------------------------------|--------------|--|--|
| | | | | | | | Type 2 | Type 3A | Type 3B |
| 28. | Obeidat et al., 2018 [42] | Private hospitals | Registered nurses | 274 (49%; 51%) | Jordan | Men were more likely to report a higher overall rate of perceived workplace bullying (p < 0.001) than women. | X | X | X |
| 29. | Orlino et al., 2022 [43] | Vascular surgery | Medical Trainees | 132 (31%; 69%); | USA | Experience of Bullying (36%; 30%) | X | X | X |
| 30. | Pinar et al., 2017 [44] | All levels healthcare institutions | Health personnel | 12,944 (60%; 40%) | Turkey | Workplace violence in 12 months (48%; 39.5%) Violence during the career (54.3%; 49.4%) | X | X | X |
| 31. | Scruggs et al., 2020 [45] | Ophthalmology | Trainees | 112 (47%; 53%) | USA | Sexual harassment from patients (86.8%; 44.1%). Physical harassment (24.5%; 8.5%) | X | X | X |
| 32. | Sharma et al., 2021 [46] | Cardiology | Cardiologists | 5931 23%; (77%) | Globally | Gender discrimination/sexual harassment (57%; 22%) Hostile work environment (68%; 37%) Sexual harassment (12%; 1%). | X | X | X |
| 33. | Siller et al., 2017 [47] | A medical university | Students | 88 (51%; 49%) | Austria | Harassment and sexual mistreatment (68.9%; 32.6%). Humiliation (77.8%; 53.5%). | | X | X |
| 34. | Simoes et al., 2020 [48] | Primary and secondary care | Nursing, medical, and other personnel | 203 (71%; 29%) | Brazil | Some form of Abuse 40.4% (48%; 22%). | X | X | X |
| 35. | Subbiah et al., 2022 [49] | Oncology | Oncologists | 271 (56%; 44%) | USA | Sexual harassment-peers and superiors (80%; 56%) Sexual harassment-patients and families (67%; 35%) | X | X | X |
| 36. | Swed et al., 2022 [50] | Graduate medical education | Medical residents and fellows | 276 (58%; 38%); missing (3.6%) | Syria | Bullying (54%; 30%) | X | X | X |
| 37. | Tekin and Bulut, 2014 [51] | Operating room | Nurses | 360 (92%; 8%) | Turkey | A significant relationship-between gender and educational status (p<0.05); women were more exposed to verbal abuse. | | X | X |
| 38. | Urnberg et al., 2022 [52] | All sectors, public and private | Physicians | 2786 (67%; 33%) | Finland | Overall aggression (73%; 27%) Physical Aggression (69%; 31%) Non-physical Aggression (77% 23%) | X | X | X |
| 39. | Zampieron et al., 2010 [53] | All levels of health care institution | Nursing personnel | 579 (79%; 21%) | Italy | Aggression (52%; 42%) Verbal aggression (82.8%; 78%) | X | X | X |
| 40. | Speroni et al., 2014 [54] | Multiple-hospital system | Nurses | 762 541 (93%; 7%) | USA | Workplace violence 76% (93%; 7%) | X | | |
| 41. | Wang et al., 2020 [55] | Vascular surgery | Vascular residents | 284 (36%; 64%) | USA | Sexually harassment (25%; 1%) GBDB during training (80%; 14%) Some form of public humiliation (64%, 49%) | X | | X |
| 42. | Weldehawaryat et al., 2020 [56] | Public health facilities | Nurses | 348 (57%; 43%) | Ethiopia | WPV (61%; 39%) | X | | X |

(Continued)

**Table 2.** (Continued)

| S. # | Author/s, year | Clinical Setting | Professional Category/ies | Sample (Female; Male) | Country/ies | Prevalence of WPV by Type (Female; Male) | Perpetrators | | |
|------|----------------|------------------|---------------------------|----------------------|-------------|------------------------------------------|--------------|--------------|--------------|
| | | | | | | | Type 2 | Type 3A | Type 3B |
| 43. | Feng et al., 2022 [57] | GPs | GPs | 4376 (59%; 41%) | China | Any type (49%; 51%) Physical (37%; 63%); Non-physical (49%; 51%) | X | | |
| 44. | Özdamar Ünal et al., 2022 [58] | Various healthcare settings | Physicians and others | 701 (68%; 32%) | Turkey | Workplace violence (70%; 30%) | X | | |
| 45. | Turgut et al., 2021 [59] | Emergency department | Physicians | 157 (37.6%; 62.4) | Turkey | Violence-reported cases (37.6%; 62.4%) | X | | |
| 46. | Vezyridis et al., 2015 [60] | emergency departments | Nursing and medical personnel and a few others | 220 (62%, 38%) | Cyprus Republic | No significant differences between the participant's gender. | X | X | |
| 47. | Elston & Gabe, 2016 [61] | Primary health care | GPs | 697 (37%; 62%) | South-east England, UK | Physical assault (7%; 13%); Threat of harm (8%; 33%) Verbal abuse (78%; 74%) Afraid of becoming a victim of violence (76%;60%) | X | | |
| 48. | Oguz et al., 2020 [62] | Pediatric clinics | Medical and nursing personnel and others | 182 (78.5; 21.5%) | Turkey | Violence (72%; 27%) | X | | |
| 49. | Newman et al., 2011 [63] | Rural and urban settings | Midwives, Nurses, Physicians | 297 (69%; 31%) | Rwanda | Verbal abuse: (68%; 32%); Bullying: (66%; 34%); Sexual harassment: (75%; 25%); Physical attacks: (64%; 36%) | X | X | X |
| 50. | Arnold et al., 2020 [64] | Pediatrics, internal medicine, and surgery Residents | Physicians | 381 (60%; 40%) | USA | Some sort of harassment during both medical school and residency (55.8%; 35.6%) Sexual harassment (83%; 44%) | X | | X |
| 51. | Benzil et al., 2020 [65] | Neurosurgery | Surgeons | 622 (21%; 78%; others 1%) | USA-professional platforms | Sexual harassment (88%;44) | | X | X |
| 52. | Freedman-Weiss et al., 2020 [66] | Surgical training programs | Trinee residents | 270 (44%; 53%) others; 3% | USA | Sexual Harassment 49% (70.8%; 30.8%) | | X | X |
| 53. | Nukala et al., 2020 [67] | Vascular surgery | Medical Trainees | 133 (37%; 61%; others 2%) | USA | Sexual harassment (52%; 23%) | | X | X |
| 54. | Smed & Aulivola 2020 [68] | Vascular surgery | Faculty of training programs | 149 (22%; 8%) | USA | Sexual harassment (67%;34%) | | X | X |
| 55. | Crebbin et al., 2015 [69] | Surgery | Medical personnel in surgery | 3516 (19%; 81%) | Australia and New Zealand | Overall prevalence of DBSH (72%; 64%) Bullying (58%; 34%); Sexual Harassment (30%; 2%) | | X | X |
| 56. | Jain et al., 2019 [70] | Ophthalmology | Ophthalmologists | 282 (32%;68%) | Australian | Bullying (43%; 33%); Discrimination (31%; 8%) Sexual harassment (23%; 0.5%) | | | X |
| 57. | Lall et al., 2021 [71] | Emergency | EM residents | 8470 (35%; 62%), Unknown (3%) | USA | Gender discrimination (65%; 9%) Verbal or emotional abuse (32%; 27%) | X | | X |
| 58. | Picakciefe et al., 2017 [72] | Primary health care | Health personnel | 119 (83%;17%) | Turkey | Mobbing 31% (89%; 11%) | | | X |
| 59. | Vorderwulbecke et al., 2015 [73] | Primary health care | Primary care physicians | 1500 (40%; 60%) | Germany | Aggression (60%; 51%); Sexual harassment (25%; 15%) | X | | |

(Continued)

**Table 2.** (Continued)

| S. # | Author/s, year | Clinical Setting | Professional Category/ies | Sample (Female; Male) | Country/ies | Prevalence of WPV by Type (Female; Male) | Perpetrators | | |
|---|---|---|---|---|---|---|---|---|---|
| | | | | | | | Type 2 | Type 3A | Type 3B |
| 60. | Xie et al., 2017 [74] | Medical school | Medical students' | 180 (56%; 445) | China | All types of violence (29%; 33%) Sexual harassment (9%; 10%) Physical violence (15%; 3%) | X | | |
| 61. | Al-Maskari et al., 2020 [75] | Emergency department | Nurses | 103 (74%; 26%) | Oman | Physical: (47% 53%) Non-physical: (24%; 76%) | X | | |
| 62. | Ferri et al., 2016 [76] | General Hospital | Nursing and medical personnel | 419 (67%;33%) | Italy | WPV (assaulted): 45% (72%; 28%) | X | | |
| 63. | James et al., 2011 [77] | Psychiatric in-patient settings | Mental health nurses | 76 (71%; 31) | Nigeria | Male nurses reported significantly higher episodes of aggressive spitting behaviour (p<0.011) as well as physical violence (p<0.010). | X | | |
| 64. | Hsiao et al., 2021 [78] | University of Florida College of Medicine | Medical personnel | 509 (54%; 46% | USA | Sexual harassment (46.2%;19.4%) | X | X | |
| 65. | Meyer et al., 2021 [79] | Ophthalmology | Trainees and Ophthalmologist | In 2015–582 (29%; 71%) In 2018–560 (29%; 71%) | Australia and New Zealand | Sexual harassment (32%; 4%). Discrimination (43%;12%) Bullying (51%; 31%) | | | X |
| 66. | Hu et al., 2022 [80] | Association of American Medical Colleges | Physicians | 6000 (29%; 66%); non-binary not included | USA | Greater representation of women within a specialty is associated with a lower prevalence of harassment experienced by men and women physicians (e.g., threats of physical harm for women (OR = 0.973, CI 0.954–0.992) and men (0.984, CI 0.974–0.993) and unwanted sexual advances for women (OR = 0.976, CI 0.967–0.984) and men (0.988, CI 0.981–0.995). | X | X | X |
| 67. | Forrest et al., 2011 [81] | All urban and rural settings | General practitioners | 804 (51%; 49%) | Australia | Verbal Abuse (56%; 58%); Physical Abuse (4%;7%), Stalking (3%;4%); Sexual Harassment (10%;2.5%) | X | | |
| 68. | Abed et al., 2016 [82] | Primary Care clinics | Nurses and physicians | 102 (86%; 14%) | Barbados | Any type of violence (71%; 21%) Verbal abuse: (67%; 21%) | X | X | |
| 69. | Vyas et al., 2022 [83] | Tertiary Care Hospital | Nurses, Physicians, and others | 157 Sample not sex-segregated | Uttarakhand North India | Overall violence (65%; 35%) Verbal violence (62%; 38%); Bullying (100%; 0%) Physical violence (60%; 40%) | X | | |
| 70. | Giglio et al., 2022 [84] | Orthopedics | Physicians | 465 (28%; 72%), NB:1 (0.2%) | Canada | Gender-based harassment (98%; 68) Sexual harassment (83%; 71) | X | X | X |
| 71. | Abrams & Robinson, 2011 [85] | Urban settings | Physicians (directory of physicians) | 1190 (35%; 61%; unknown 4%) | Canadian urban area | Stalking (13.5%; 10.9%) | X | | |
| 72. | Afkhamzadeh, 2019 [86] | Teaching hospital | Physicians & medical students | 321 (55%; 45%) | Iran | At least one type of violence (51%; 68%) | X | | |

(Continued)

**Table 2.** (Continued)

| S. # | Author/s, year | Clinical Setting | Professional Category/ies | Sample (Female; Male) | Country/ies | Prevalence of WPV by Type (Female; Male) | Perpetrators | | |
|---|---|---|---|---|---|---|---|---|---|
| | | | | | | | Type 2 | Type 3A | Type 3B |
| 73. | Alhamad et al., 2021 [87] | All kinds of hospitals | Physicians | 969 (35%; 65%) | Jordan | Abuse: (55%; 67%) Verbal abuse: (50%; 60%); Physical abuse: (01%; 08%) | X | X | |
| 74. | Al. Shamlan et la., 2017 [88] | Teaching Hospital | Nurses | 391 (89%; 11%) | Saudi Arabia | Verbal abuse (28%; 50%) | X | X | |
| 75. | Balch Samora et al., 2020 [89] | Orthopedics | Surgeons | 926 (67%; 33%) | USA | Overall DBSH: (81%; 35%); Discrimination (84%; 59%); Sexual harassment (54%; 10%) | | | X |
| 76. | Bhandari et al., 2021 [90] | Internal medicine | Hospitalists | 336 (57%;43%) | USA | Discrimination (99%; 29%) Harassment (72%; 36%, | X | X | |
| 77. | Jaradat et al., 2016 [91] | Hospitals and primary care clinics | Nurses | 343 (62%; 38%) | State of Palestine | Workplace Aggression (26%; 28%) Physical aggression (5%; 5%) Verbal Aggression 24%; 25%); Bullying (5%; 12%) | X | X | |
| 78. | Cashmore et al., 2012 [92] | Correctional health services | Physicians and nurses, and others | 208 incidents | Australia | Workplace violence incidents (66%; 34%) Verbal Abuse (74%, 26%); Physical Abuse (25%; 55) | X | | |
| 79. | Chen et al., 2018 [93] | Tertiary teaching hospital | Nurses | 1983 (92%; 8%) | China | Any type of violence (50%; 38%) Non-physical (50%; 47%); Physical (6%; 9%) | X | X | |
| 80. | Cheng et al., 2020 [94] | National Health Services | Nurses, Midwives | 147 (82%; 18%) | England, UK | Aggression from Patients (26%; 20%) | X | X | |
| 81. | Dal Pai et al., 2015 [95] | Hospital setting | Medical, nursing, and other health personnel | 269 (58%; 42%) | Brazil | Exposed to Violence (71%; 52%) | X | | |
| 82. | Dehghan-chaloshtari and Ghoduosi, 2017 [96] | Hospital | Nurses | 100 (76%; 24%) | Iran | Physical (82.5%; 17.5%); Verbal violence (78.6%; 21.4%) Bullying and mobbing (70.3%; 29.7%) Sexual abuse (66.7%; 33.3%) | X | X | X |
| 83. | Demeur et al., 2018 [97] | Flemish (Belgian Federal State) | GPs | 248 (60%; 40%) | Belgium | Raising voice (71%; 29%); Scolding (63%; 37%) Verbal intimidation (63%; 37%) Violating privacy (60.5%; 39.5%); Touching (63%, 37%) Grabbing, slapping & kicking (62%; 38%) Sexual intimidation (70%; 30%) | X | | |
| 84. | Difazio et al., 2019 [98] | Diverse healthcare settings | Members of the Russian Nurses Association | 438 (97.5%; 2.5%) | Russian Federation | Bullying (97.5%; 2.5%) | | X | |
| 85. | Fitzgerald et al., 2019 [99] | Academic teaching hospitals | Surgical Residents | 76 (49%; 51%) | USA | At least one form of abuse and harassment (48%; 52.5%). Discrimination in relation to gender (92%; 8%). | X | | X |
| 86. | Fnais et al., 2013 [100] | Residency programs in teaching hospitals | Trainee residents | 213 (42%; 58%) | Saudi Arabia | Verbal harassment during training (76%; 51%) Gender discrimination (69%; 57%) Sexual harassment (28%; 13%) | X | | X |

(Continued)

**Table 2.** (Continued)

| S. # | Author/s, year | Clinical Setting | Professional Category/ies | Sample (Female; Male) | Country/ies | Prevalence of WPV by Type (Female; Male) | Perpetrators | | |
|------|----------------|------------------|---------------------------|----------------------|-------------|------------------------------------------|--------|---------|---------|
| | | | | | | | Type 2 | Type 3A | Type 3B |
| 87. | Fute et al., 2015 [101] | Public health facilities | Nurses | 642 (63%; 37%) | Ethiopia | Workplace violence (36%; 20%) | X | | |
| 88. | Hills et al., 2012 [102] | Clinical medical practitioners | Medical personnel | 9438 (43%; 57%) | Australia | Verbal or written aggression (72.6%; 69%) Physical aggression (33.8%; 31.2%) | X | X | |
| 89. | Honarvar et al., 2019 [103] | University-affiliated public hospitals | Nursing personnel | 405 (81%; 19%) | Iran | Verbal Abuse (84%; 83%); Verbal threat (25.6%; 46.4%) Physical violence (16.8%; 41.6%) Sexual Harassment (9.8%; 15.6%) | X | | |
| 90. | Kisiel et al., 2020 [104] | Uppsala University | Medical students | **2002**–343 (55%: 45%) **2013**–720 (62%; 38%) | Sweden | **2013** (pre-clinical group) Discrimination (22%; 15%); Favoritism (23%; 18%) Intrusive, unwelcome acts (21%; 15%) **2013** (clinical group) Discrimination (41% and 25%); Favoritism (53%; 33%) Intrusive, unwelcome acts (26%; 20%) | | | X |
| 91. | Lafta & Falah, 2019 [105] | Hospitals and primary healthcare centres | Medical, nursing personnel and others. | 700 (51%; 49%) | Iraq | Physical violence (24%; 76%) Verbal (53%; 47%) | X | | |
| 92. | Li et al., 2020 [106] | General Hospitals Survey | Nursing Workforce | 396 (73%; 27%) | China | Verbal abuse from patients and/families (34%;30%) Verbal abuse from staff (59%; 47%) Physical abuse from patients/families (64%; 58%) Physical abuse from staff (81%; 70%) | X | X | |
| 93. | Li et al., 2010 [107] | Emergency Medicine | Residents | 196 (53%; 47%) | USA | Sexual harassment 23% (37%; 8%). | X | | |
| 94. | Lu et al., 2020 [108] | Academic Emergency Medicine faculty | Emergency Medicine faculty | 144 (39%; 61%) | England, UK | Discrimination based on gender (62.7%; 12.5%) Unwanted sexual harassment behaviors (52.9%; 26.2% | X | | X |
| 95. | Lucas-Guerrero et al., 2020 [109] | General Surgery | Surgical residents | 452 (66%; 34%) | Spain | Physical abuse (5.4%; 10.5%) Sexual harassment (21.4%; 6.5%) Discrimination (90.4%, 9.6%). | X | | X |
| 96. | Margavi et al., 2020 [110] | Emergency wards of teaching hospitals | Nurses | 140 (61%; 39%) | Iran | Physical violence (43%; 76%) Psychological violence (90%; 83%) Sexual harassment (8%; 0%) Bullying/mobbing (34%; 46%) | X | | |
| 97. | McKinley et al., 2019 [111] | Residency training programs | Medical residents | 371 (46%; 53%); Others 2) | Massachusetts, USA | Gender-based discrimination (93%; 24%). Sexual harassment during training (34%; 5) | X | | X |
| 98. | Mirza et al., 2012 [112] | Emergency department | Physicians in training | 675 (47%; 53%) | Pakistan | Verbal abuse (61; 63%) Physical abuse (8%; 15%) | X | | |

(Continued)

**Table 2.** (Continued)

| S. # | Author/s, year | Clinical Setting | Professional Category/ies | Sample (Female; Male) | Country/ies | Prevalence of WPV by Type (Female; Male) | Type 2 | Type 3A | Type 3B |
|---|---|---|---|---|---|---|---|---|---|
| 99. | Moman et al., 2020 [113] | Pain management clinicians | Medical and nursing personnel | 58 (41%; 59%) | Conference participants, USA | Experienced assault (59%; 75%) | X | | |
| 100. | Moylan et al., 2014 [114] | Psychiatric facilities | Nurses | 110 (85%; 15%) | New York, USA | Physical assault 73% (80%; 20%) | X | | |
| 101. | Nieto-Gutierrez et al., 2018 [115] | Residency programs | Medical residents | 1054 (38%; 62%) | Peru, South America | Workplace Violence (75%; 72%) | X | X | |
| 102. | Park & Choi, 2020 [116] | General Hospital | Nurses | 205 (89%; 11%) | South Korea | Mean experience of Verbal violence (23, 29) Mean Doing Verbal violence (19, 24) | | | X |
| 103. | Pol et al., 2019 [117] | Intensive Care Unit | ICU Nurse clinicians | 47 Patient records | Australia | Verbal violence (20%; 66.7%) Physical violence (45.7%; 25%) | X | | |
| 104. | Prajapati et al., 2013 [118] | All kinds of health facilities | Medical, nursing, midwifery, and other personnel | 747 | Nepal | Gender-based harassment (62.5%; 37.5%) Sexual Harassment (56.5%; 43.5%) | X | | |
| 105. | Rosta & Aasland, 2018 [119] | Hospital | Medical personnel | **1993** = 2439 (28%; 72%) **2004** = 730 (31.5%; 68.5) **2014–15** = 1080 (43%; 67%) | Norway | Perceived Bullying in: 1993 = (8.3%; 4.7%) 2004 = (4.8%; 8.4%) 2014–15 = (9.2%;5.4%) | | X | X |
| 106. | Rouse et al., 2016 [120] | Academic settings | Family physicians | 1065 (43%; 57%) | USA | Ever displayed bullying behaviors (7.7%; 11.2%) Ever been bullied (34%; 24.7) | | X | |
| 107. | Sachdeva et al., 2019 [121] | Emergency Department | Medical and nursing personnel | 335 (34%; 66%) | India | Verbal abuse (37%; 63%); Physical abuse (32%; 68%) Confrontation (12%;88%) | X | X | |
| 108. | Sakellaropoulos et al., 2011 [122] | Anesthesia | Certified registered nurse anesthetists | 205 (62%; 37%) | USA | Verbal aggression (89%; 83%) | | X | X |
| 109. | Tian et al., 2020 [123] | Various Hospitals | Nursing and medical personnel | 3684 (85%; 15%) | China | Emotional abuse (47.3%; 55.4%); Threats (25%; 38%) Physical Abuse (14.5%; 24%); Sexual Abuse (7%; 13%) | X | | |
| 110. | Vargas et al., 2020 [124] | University Medical School | Faculty members | 705 (48; 52%) | USA | Sexual harassment from insiders (82.5%; 65.1%) Sexual harassment from patients (64.4%; 44.1%) Gender harassment from insiders (82.2%; 64.9%) Gender harassment from patients (64.0%; 44.1%) | X | X | X |
| 111. | Viottini et al., 2020 [125] | University Hospital Network | Midwives, nurses and physicians | 364 (77.5%; 22.5) | Italy | Assaults incidences (77.5%; 18.5%) | X | X | |
| 112. | Williams et al., 2021 [126] | Medical residency program | Internal Medicine residents | 33 (41%; 59.3%) | USA | Microaggression by a patient (90.9%; 56.3%) | X | | |
| 113. | Camm et al., 2022 [127] | Cardiology | Trainees | 1359 (27%; 73%) | UK | Sexist language (14%; 4%) | | | X |
| 114. | Rowe et al., 2022 [128] | Large academic medical center | Physicians | 1505 (49%; 42%); unknown: 143 (9%) | California, USA | Any forms of mistreatment (31.0%; 15.0%) Sexual harassment (8.8%;1.5%) Verbal abuse (28%; 14%); Physical abuse (6%; 4%) | X | X | |

(Continued)

**Table 2.** (Continued)

| S. # | Author/s, year | Clinical Setting | Professional Category/ies | Sample (Female; Male) | Country/ies | Prevalence of WPV by Type (Female; Male) | Perpetrators | | |
|------|---------------|------------------|--------------------------|----------------------|-------------|------------------------------------------|--------|--------|--------|
| | | | | | | | Type 2 | Type 3A | Type 3B |
| 115. | Vidal-Alves et al., 2021 [129] | Public hospitals | Nursing staff | 950 (78%; 22%) | Southeast of Spain | Mean- personal lateral violence (3.06; 3.41) Mean- social lateral violence (1.92; 1.57) Mean -work-related lateral violence (1.51; 1.28) | | X | |
| 116. | Kibunja et al., 2021 [130] | Emergency department | Nurses | 82 (65%; 35%) | Kenya | Physical violence (71%; 29%); Verbal abuse (65%; 35%); Sexual harassment (82%; 18%) | X | | |
| 117. | Sabak et al., 2021 [131] | ED | Physicians | 362 (38%;62%) | Turkey | Verbal threats (100%; 97%); Sexual harassment (5%;7%) Physical assaults (50%; 57%); Stalking (16%; 30%) | X | | |
| 118. | Gadjradj et al. 2021 [132] | Neurology | Neurosurgeons and neurosurgical residents | 503 (20%80%) | e-survey for conference participants | Gender discrimination (90.2%; 13.0%) | X | X | X |
| 119. | Albuainain et al., 2022 [133] | Surgical environments | Physicians | 788 (35%; 65%) | Saudi Arabia | Negative Attitude Questionnaire-R score (42.7; 42.3) | | X | X |
| 120. | Nøland et al. 2021 [134] | 2-cohorts of medical students | Physicians | 893 (56%; 43%) | Norway | Prevalence of WPV T2 = (14.5%; 27.7%); T3 (11.3%; 25.0%); T4 (9.1%; 14.4%); T5 (7.3%; 10.5%) | X | | X |
| 121. | Papantoniou, 2022 [135] | Greek NHS | Nurses | 1726 (71%; 29%) | Greece | Sexual harassment (67%; 41%). | | | X |
| 122. | Kowalczuk & Krajewska-Kulak, 2017 [136] | General Hospital | Medical, nursing, midwifery, and other personnel | 1624 Medical (56%; 44%) Nurses (98%; 2%) Midwives (99.6%; 0.4%) | Poland | Mean scores of patient aggression 0–5 scale Nurses (26.6; 34.9) Physicians (17.8; 19.7) Midwives (12; 10.9) | X | | |
| 123. | Notaro et al., 2021 [137] | Dermatology | Dermatologists and trainees. | 330 (75%; 24%, 1%-unknown) | USA | Sexual harassment (94%; 52%) Sexual assaults (35%; 15%) | X | | |
| 124. | Schlick et al., 2021 [138] | 301 general surgery programs | Medical residents | 6956 (41%; 59%) | USA | Gender discrimination (80%; 17%) Sexual harassment (43%; 22%) | X | X | X |
| 125. | Hunter et al., 2022 [139] | Higher Education Institution | Nursing students | 138 (92%; 8%) | Scotland, UK | Ever experienced Verbal violence (70%; 67%). Ever experienced Physical violence (72%; 41%) | X | | |
| 126. | Ferrara et al., 2021 [140] | Academic | Nursing students | 603 (77%; 23%) | Italy | Psychological violence (39%; 22%) Physical violence (9%; 5%) | X | X | X |
| 127. | Pendleton et al., 2021 [141] | Three academic institutions | Medical residents from 12 programs | 309 (55%; 45%) | Boston, USA | Gender-based discrimination (100%; 69%) | X | | |
| 128. | Snavely et al., 2021 [142] | Emergency medicine (EM) | Resident trainees | 22 (64%; 36%) | New York, USA | Mean number of incidents/shift (3.0; 0.9) | X | | |

(Continued)

**Table 2.** (Continued)

| S. # | Author/s, year | Clinical Setting | Professional Category/ies | Sample (Female; Male) | Country/ies | Prevalence of WPV by Type (Female; Male) | Perpetrators | | |
|------|----------------|------------------|---------------------------|------------------------|-------------|-------------------------------------------|---------------|---|---|
| | | | | | | | Type 2 | Type 3A | Type 3B |
| 129. | Hock et al, 2021 [143] | ophthalmology, surgery, medicine and others | Medical residents and faculty | 91 (50%; 48%; No response 2%) | Iowa, USA | Recognition for patient-initiated harassment on a 5-point Likert scale before the workshop (4.0; 3.7) and after workshop participation (4.6; 4.5) | X | | |
| 130. | Rodriguez-Acosta et al., 2010 [144] | University Hospitals | Nursing staff | 220 injuries (86%; 14%) | Duke, North Carolina, USA | While the number of assaults was greater among women than men, their risk was lower (RR = 0.70). | X | | |
| 131. | Hahn et al., 2013 [145] | University general hospital settings | Physicians, nurses and midwives | 2495 (82%; 18%) | Switzerland | Gender factored WPV (no clear description provided) | X | | |
| 132. | Wang et al., 2022 [146] | ICU | Nurses | 305 (68%; 32%) | China | There were increased odds of experiencing WPV among nurses with lower professional titles, male nurses (OR = 2.7, CI = 1.310 to 5.944), and those with less than five years of experience. | X | | |
| 133. | Wright & Khatri, 2015 [147] | Teaching hospital network | Nurses | 1078 (91%; 9%) | USA | Male nurses experienced significantly higher work-related bullying than female nurses p < .067). No significant differences in person-related bullying, which had a significant positive relationship with both psychological/behavioral responses and medical errors. | | | X |
| 134. | Bambi et al., 2014 [148] | Prehospital EMS, emergency department, ICU, and OR | Nurses | 1202 (61.5%; 38.5%) | Italy | Desire to leave the nursing profession because of the LH was (15.5% and 9%); however, gender was not statistically significant for LH. | | X | |
| 135. | Koukia et al., 2014 [149] | General Hospital | Healthcare staff, including nursing and medical personnel | 250 (74%;26%) | Greece | Women were more likely to experience sexual (p<0.012) and physical violence (p<0.014). | | X | X |
| 136. | al-Omari, 2015 [150] | 11 General hospitals | Nurses | 468 (47%; 53%) | Jordan | Female nurses were 0.5 times less likely to report being physically attacked than male nurses (p = 0.003). Female nurses were 1.5 times more likely to report being verbally abused than male nurses (p = 0.046). | X | | |
| 137. | Esmaeilpour et al., 2011 [151] | Emergency department | Nurses | 186 (89%; 11%) | Iran | Male nurses were the victims of physical violence more often than female nurses (p = 0.000). | X | | |
| 138. | Joa and Morken, 2012 [152] | Out-of-Hours primary care centres | Physicians, nurses, and others | 536 (70%; 30%) | Norway | Men were more at risk of physical abuse (OR = 2.36, CI 1.11–5.05) and verbal abuse (OR = 1.23, 0.68–2.18). | X | | |

(Continued)

**Table 2.** (Continued)

| S. # | Author/s, year | Clinical Setting | Professional Category/ies | Sample (Female; Male) | Country/ies | Prevalence of WPV by Type (Female; Male) | Perpetrators | | |
|---|---|---|---|---|---|---|---|---|---|
| | | | | | | | Type 2 | Type 3A | Type 3B |
| 139. | Serafin & Czarkowska-Pączek, 2019 [153] | Polish healthcare facilities | Nursing personnel | 411 (96%; 4%) | Poland | Women were more often affected by 'being humiliated or ridiculed in connection with their work' (p = 0.040), 'being ordered to do work below their level of competence' (p = 0.010), and 'having key areas of responsibility removed or replaced with more unpleasant tasks' (p = 0.005). | | | X |
| 140. | Kelly et al., 2015 [154] | Forensic hospital | Overall staff | 488 (69%; 31%) | California, USA | Men experienced higher scaled frequencies of assault than women (4-point Likert scale, 0–3] mean = (0.46 vs 0.33, p = 0.02). | X | | |
| 141. | Fafliora et al., 2015 [155] | Primary, secondary, and tertiary care hospital | Nurses | 80 (83%;17% | Greece | Men (OR, 0.08, CI 0.01–0.56) and higher experience nurses (OR, 0.82, CI 0.70–0.097) were less affected by WPV. | X | X | |
| 142. | Askew et al., 2012 [156] | All doctors at Australian Medical Board | Doctors from various department | 747 (53%; 47%) | Australia | No significant differences in the prevalence of bullying between the sexes. Victims of bullying had poorer mental health (p<0.001) | | | X |
| 143. | Lindquist et al., 2020 [157] | National emergency medicine conference | Physicians and medical students | 63 (56%; 41%; Missing: 3%) | Myanmar | Women were more likely to experience verbal assault (OR = 1.18, 0.42–3.33). | X | X | |
| 144. | Yohe et al., 2020 [158] | Orthopedic residency | Orthopedic residents | 1207 (17%; 83%) | USA | Gender was not statistically significant (OR = 1.07, 0.63–1.84, p-0.79). | X | | X |
| 145. | Farid et al., 2021 [159] | Obstetrics and gynecology | Physicians | 87 (75%; 25%) | USA | Most physicians (71%) had ever experienced discrimination-attributed to their gender. | X | X | |
| 146. | Vingers 2018 [160] | Nursing Education | Nursing students | 107 (87%; 13%) | USA | There was no significant difference in the frequencies of reported bullying behaviors for male and female nursing students (< .05 level). | X | | X |
| 147. | Dafny& Beccaria, 2020 [161] | Three regional hospitals | Nurses | 23 (74%; 26%) | Australia | Five themes identified including, perpetrators of violence, and gender and the incidents of violence | X | | |
| 148. | Ko & Dorri, 2019 [162] | Primary care | Clinicians (nurses & Physicians) | 26 (62%; 38%) | USA | Participants reported experiences of bias, harassment, and discrimination based on gender, race/ethnicity, and/or sexual orientation and gender identity. | | X | |

(Continued)

**Table 2.** (Continued)

| S. # | Author/s, year | Clinical Setting | Professional Category/ies | Sample (Female; Male) | Country/ies | Prevalence of WPV by Type (Female; Male) | Perpetrators | | |
|---|---|---|---|---|---|---|---|---|---|
| | | | | | | | Type 2 | Type 3A | Type 3B |
| 149. | Neiterman & Bourgeault, 2015 [163] | IMGs and IENs | Medical and nursing personnel | 140 IMG 69 (36%;11%) IENs 71 (60%; 11%) | Canada | Physicians were concerned with instances of discrimination within their own professional group. Nurses, on the other hand, reported discrimination at the hands of patients and their families as well as racialization by physicians, management, and other nurses. | X | X | X |
| 150. | Al Khatib et al., 2023 [164] | ED- Public hospitals | Physicians and nurses | 163 (27%; 73%) | Jordan | Physical (2.3%; 43.7%); Verbal (29.5%; 61.3%) | X | X | X |
| 151. | Al-Wathinani et al., 2023 [165] | ED | Physicians, nurses and others | 206 (43.2%; 56.8%) | Saudi Arabia | Physical assault (48.3%; 66.7%) | X | | |
| 152. | Alhassan et al., 2023 [166] | ED | Midwives, nurses, physicians and others | 7398 (48.7%; 51.3%) | Saudi Arabia | Physical attacks (44%; 56%) | X | | |
| 153. | AlHassan et al., 2023 [167] | ED | Midwives, nurses, physicians and others | 7398 (48.7%; 51.3%) | Saudi Arabia | Sexual attack (61%; 39%) | X | X | |
| 154. | Banga et al., 2023 [168] | Health workers | Nurses, physicians and others | 5405 (53%; 45%; others 2%) | 79 countries | Verbal violence (50.8%; 51%); Physical abuse (19%; 24.2%); Emotional violence (30.6%; 28%); Sexual violence (7.4%; 3.8%) | X | X | X |
| 155. | Barequet et al., 2023 [169] | Ophthalmology | Physicians | 252 (46%; 54%) | Israel | Physical Abuse (18%;13%); Sexual harassment (50%;13%) | X | X | |
| 156. | Bekalu et al., 2023 [170] | Public hospitals | Nurses | 534 (45%; 55%) | Northeast Ethiopia | Violence (58%; 42%) | X | X | |
| 157. | Crombie et al., 2024 [171] | University | Medical students | 443 (73%; 27%) | South African | Mistreatment (80.9%; 70.8%) | | X | X |
| 158. | DiFiori et al., 2023 [172] | Orthopaedic surgery | Surgeons, fellow and residents | 105 (12%; 84%; others 2.4%), no response 2.4 | USA | Demographic information including sex did not have a statistically significant (p-0.167) association with self-reported bullying | X | X | |
| 159. | Domínguez et al., 2023 [173] | Surgery | General surgery residents | 302 (42%; 58%) | Colombia | Occasional bullying (26%; 29%); Continuous bullying (22%;21%); Sexual harassment (29.1%; 4.55) | | X | X |
| 160. | Ebrahim et al., 2023 [174] | Healthcare workers-Hospital | Nurses, physicians and others | 5537 | Kenya | Discrimination based on gender (27.3%; 20.2%) Physical abuse (5.4%; 3.8%); Verbal abuse (60.9%; 48.9%) | X | X | X |
| 161. | Forsythe et al., 2023 [175] | Vascular diseases physicians | Consultant, fellows, Residents, interns/students | 587 (35.8%; 62.9%; other 1.5%) | International (28 countries) | Experiences of Bullying, undermining behaviour, and harassment (53%; 38%) | X | X | X |
| 162. | Grover et al., 2023 [176] | General surgery and urology | Residents | 23 (35%;65%) | Mid-Atlantic | Mistreatment (88%; 33%); Verbal assault (50%; 33%) | X | | |
| 163. | Ioanidis et al., 2023 [177] | Otolaryngology | Residents and attending Physicians | 183 (55.6%; 45.4%) | Canada | **Harassment during residency** (75.2%;37.9%) Verbal (87%; 92%); Sexual harassment (45%; 11%) **Harassment- attending physicians** (68%; 31.6%) Verbal (93%; 65%); Sexual harassment (41%; 22%) | **X** | **X** | **X** |

(Continued)

**Table 2.** (*Continued*)

| S. # | Author/s, year | Clinical Setting | Professional Category/ies | Sample (Female; Male) | Country/ies | Prevalence of WPV by Type (Female; Male) | Perpetrators | | |
|---|---|---|---|---|---|---|---|---|---|
| | | | | | | | Type 2 | Type 3A | Type 3B |
| 164. | Janatolmakan et al., 2023 [178] | Emergency | Hospital nurses | 150 (58.7%; 41.3%) | Iran | Physical Violence (60.2%; 39.8%) Verbal Violence (58.5%; 41.5%) | X | X | X |
| 165. | Meese et al., 2024 [179] | Healthcare | Healthcare workers, including Nurses | 2659 (60.5%; 39.5%) | USA | Verbal Mistreatment (26.5%; 19.3%) Physical Violence (17.1%; 10.7%) | X | | |
| 166. | Nam et al., 2023 [180] | Departments of Urology and Obgyn at the University | Clinicians | 128 (76%; 24%) | Michigan | **Urology:** Unwanted sexual attention (68.8%; 22.7%) Gender harassment (84.4%; 40.9%); Sexual Coercion (15.6%; 0%) **OBGYN**; Unwanted sexual attention (68.8%; 54.6%) Gender harassment (84.4%; 68.2%); Sexual Coercion (3.1%; 0%) | X | | |
| 167. | Parodi et al., 2023 [181] | Health sector | Physicians and nurses | 3056 (57%; 43%) | Latin America | WPV (65.8%; 50.4%); Verbal violence (97%; 97%) Physical Violence (10%; 9%) | X | | |
| 168. | Rashid et al., 2023 [182] | Cardiology departments | Junior physicians | 1852 (43%; 57%) | Pakistan | Bullying (13.4%; 10.2%) | | | X |
| 169. | Ryan et al., 2023 [183] | Academic orthopedics | Nurses, residents/fellows, and physicians | 173 (65%; 45%) | USA | Harassment (all staffs): (27%; 24%) Harassment (residents and faculty) (46%; 24%) | X | | |
| 170. | Santosa et al., 2023 [184] | Academic surgery programs | Faculty and residents | Residents/fellows: 143 (58%; 42%) Faculty: 183 (41%; 59%) | USA | Incivility experiences among surgeons (77%; 6%) | X | X | X |
| 171. | Shahjalal et al., 2023 [185] | Tertiary care hospitals | Physicians | 406 (49%; 51%) | Bangladesh | Physical Violence 36%; 64%) | X | | |
| 172. | Tavolacci et al., 2023 [186] | Health Campus & nursing school | Midwifery, nursing and medicine students | 1152 (82.6%; 17%; others: 0.4%) | France | GBV (93.7%; 5.4%) | X | X | |
| 173. | Veronesi et al., 2023 [187] | Two public referral hospitals | Nurses, physicians and others | 7982 (74.7%; 25.3%) | Italy | WPV (65.7%; 34.3%); Physical violence (47.7%; 67.2%); Verbal (92.3%; 88.1%) | X | X | X |
| 1.74 | Vu et al., 2023 [188] | University | Medical students | 550 (75%; 25%) | Vietnam | Physical violence (10%; 22.5%) ; Verbal Violence 25.7%; 29.7%) ; Sexual violence (4.6%; 5.8%) | X | X | X |
| 175. | Yan et al., 2023 [189] | Emergency | Physicians | 14848 (29.5%; 70.5%) | China. | Any Type of violence (87.7%; 91.6%) Physical (37%; 57.5%); Non-physical (87.2%; 91.2%) | X | | |
| 176. | Jaradat et al., 2016 [190] | Hospitals and primary care clinics | Nurses | 341 (62%; 38) | State of Palestine | There were no significant differences between sex and workplace aggression resulting in psychosomatic symptoms (raged from 0.04–0.09). | X | X | |

(*Continued*)

**Table 2.** (Continued)

| S. # | Author/s, year | Clinical Setting | Professional Category/ies | Sample (Female; Male) | Country/ies | Prevalence of WPV by Type (Female; Male) | Perpetrators | | |
|------|----------------|------------------|---------------------------|-----------------------|-------------|------------------------------------------|--------|--------|--------|
| | | | | | | | Type 2 | Type 3A | Type 3B |
| 177. | Li et al., 2021 [191] | Emergency department | Emergency room nurses | 132 (91%; 9%) | Taiwan | Mental violence (54%; 50%) Physical violence (12.5%; 8.3%) | X | X | X |
| 178. | Zachariadou et al., 2018 [192] | Primary Health care clinics and general hospitals | Medical, Nursing, and other personnel | 167 (71%;29%) | Cyprus | At least one mobbing behavior (49%; 35.7%) | | | X |

## Methods

### Protocol registration and study design

Following the Joanna Briggs Institute (JBI) revised guidelines, we conducted a scoping review. The protocol for this review was registered on the Open Science Framework on January 14, 2022, and is accessible at https://osf.io/t4pfb/ and S2 Text: Registered Protocol. The scoping literature review design addressed the research questions and accommodated the heterogeneous and complex nature of the literature. This method is appropriate for exploring the extent of the literature, mapping and summarizing the evidence, and identifying and analyzing knowledge gaps to inform future research. The framework used for this review consists of eight steps; they are built upon the seminal framework of Arksey and O'Malley's scoping review, which was further developed by Levac and colleagues. The revised guidelines of JBI align these eight steps with the Preferred Reporting Items for Systematic and Meta-Analyses extension for Scoping Reviews (PRISMA-ScR), ensuring rigour, transparency, and trustworthiness in Reporting the conduct of the scoping review. The first step of the scoping review framework is to align with research objectives, the title, and the inclusion criteria, as well as the exclusion criteria (see Box 1). Please see S1 Checklist: PRISMA-ScR Checklist.

---

### Box 1. Study selection criteria.

**Inclusion Criteria for Studies**

1. The study participants included nurses and/ or physicians who experienced WPV during their careers.

2. Provided sex-segregated data for any form of violence and any type of perpetrators among nurses and physicians, including students, globally.

3. Published in English and after 2010.

**Exclusion criteria**

4. Studies that did not provide sex-segregated data and information for perpetrators.

5. Exclude systematic/ scoping reviews, concept or theoretical papers, and theses.

---

### Search strategy

The research team collaborated with a health sciences librarian to develop a comprehensive search strategy. The systematic search focused on published literature in various databases,

including Ovid MEDLINE: Epub Ahead of Print, In-Process and Other Non-Indexed Citations, which were translated in CINAHL Plus, APA PsycINFO, Web of Sciences, and Gender Studies Databases, Applied Social Sciences Index & Abstracts (ASSIA), and Sociological Abstracts (S3 Text: Ovid MEDLINE search strategy, which was translated in all other databases). The search terms related to the population (midwifery, nursing, and physicians), concepts (violence and gender-based violence), and context (healthcare) were combined appropriately based on the scoping review objectives. These terms were identified through a preliminary literature search on various aspects of workplace violence in Google Scholar. The final search results were exported to EndNote, a citation manager, to de-duplicate sources from multiple databases. After de-duplication, the sources were imported into the Covidence online software program that streamlined the screening process by two independent reviewers. The final search of the literature review for this study was conducted on 11ᵗʰ February 2024.

### Evidence screening and selection

The identified sources were screened based on the inclusion criteria (S4 Text: Excluded Sources). Two independent reviewers screened the titles and abstracts to shortlist the sources. Discrepancies were resolved through discussion and consensus, with a complete source review conducted, if necessary, followed by a full-text review against the inclusion criteria by two reviewers. The selection process is presented in the PRISMA diagram (Fig 1). Given the overall objective of the review to map the most frequent forms and prevalence of GB-WPV for midwives, nurses, and physicians in different contexts and clinical settings, a quality assessment of the identified sources was not conducted.

### Data analysis and synthesis of results

This paper is a component of a multi-part scoping review; it reports on the perpetrators of WPV from gender-segregated prevalence data reported from a global context among the health workforce, including nurses and physicians. The prevalence and risk factors have been reported elsewhere [4]. This paper reports on Type II and Type III (vertical and horizontal) WPV perpetrators. Data from all sources (S1 Data) that reported sex/gender segregated findings and provided information for the types of perpetrators were included in mapping the prevalence of GB-WPV (See Table 2) for several types/forms of WPV and the clinical setting across countries/special regions. We could not calculate a mean score for various forms of violence based on gender for all the studies that provided information on perpetrators because of the wide variability in the operational definitions of the terms and the concepts in these studies. These studies also did not consistently provide quantifiable data for the Types of perpetrators. Only 34 studies (19%) provided the gender of perpetrators. We summarized the proportion of male and female perpetrators in those studies for Type II, Type III-A (horizontal) and Type III-B (Vertical) violence (see Table 3).

### Results

After de-duplication, 8435 possible references were imported for screening in the Covidence. These studies were screened against the title by one person, 1551 were shortlisted to be screened (for title and abstract) by two independent reviewers, and 402 were assessed for full-text eligibility. After applying the inclusion and exclusion criteria, 178 [6–9, 15, 20–125] studies were retained (PRISMA diagram, Fig 1) and analyzed to report on perpetrators that provided gender-segregated findings for WPV and information on various types of perpetrators (Table 2). We included studies published between 2010–2024. The most common study design

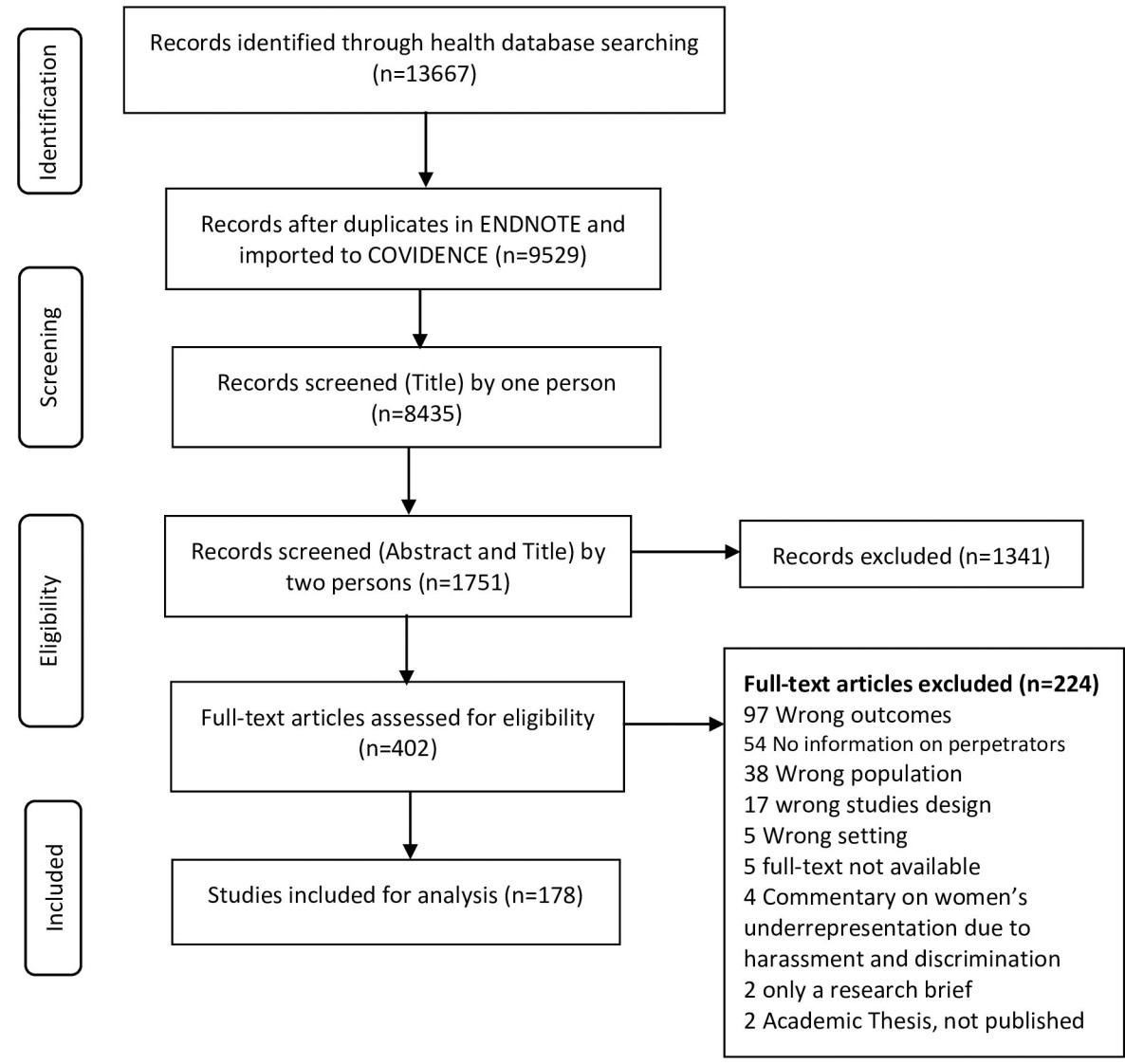

**Fig 1. PRISMA flow diagram for screening and selection.**

was quantitative, cross-sectional (n = 168), mixed methods (n = 4), and qualitative methods (n = 6).

## Perpetrators for the three types of violence

A total of 178 studies provided information on the perpetrators of either Type II (consumers/patients, including patients' companions), Type III-A (from colleagues), and Type III-B (from administrators and superior authorities within and between professions) violence. Studies included in this review did not consistently provide data for all types of violence and perpetrators; instead, they provided data for any Type/s. Of 178 studies, 141 (79%) reported perpetrators for Type II violence, followed by 93 (52.2%) for Type III-B (vertical) and 92 (51.6%) for Type III-A (horizontal) violence. Only 40 (22.5%) studies [9, 21–53, 164, 168, 172, 177, 178, 191] reported information about all three types of violence.

**Table 3. Gender and the perpetuation of workplace violence among physicians and nurses.**

| S. No | Author/s | Population | Form of Violence | Type of Perpetrators | | | Gender of Perpetrators | | |
|---|---|---|---|---|---|---|---|---|---|
| | | | | Consumers | Horizontal | Vertical | Men % | Women % | Both Genders % |
| 1. | Aghajanloo et al., 2011 [6] | Nursing students | Aggression | X | | | 66.70 | 33.30 | 0.00 |
| | | | Verbal abuse | | | X | 49.95 | 50.05 | 0.00 |
| 2. | Chang et al., 2020 [8] | Nursing students | Sexual harassment | X | | X | 88.30 | 10.40 | 1.00 |
| 3. | Al-Ghabeesh & Qattom, 2019 [15] | Nurses | Bullying | | | X | 52.50 | 17.50 | 30.00 |
| 4. | Chatziionnidis et al., 2018 [29] | Nurses and physicians | Bullying | X | X | X | 10.50 | 37.90 | 51.60 |
| 5. | Ferri et al., 2020 [35] | Nurses | Verbal abuse | X | | | 87.50 | 12.50 | 0.00 |
| | | | Verbal & Physical violence | X | | | 100.00 | 0.00 | 0.00 |
| 6. | Harthi et al., 2020 [37] | Nursing and medical personnel | Workplace Violence | X | X | X | 39.42 | 20.33 | 40.25 |
| 7. | Lei et al., 2022 [40] | Nurses | Workplace Violence | X | X | X | 83.00 | 17.00 | 0.00 |
| 8. | Menhaji et al., 2022 [41] | OB/Gyn trainees | Sexual Harassment | X | X | X | 74.30 | 10.50 | 15.20 |
| 9. | Orlino et al., 2022 [43] | Medical trainees | Bullying | X | X | X | 47.60 | 16.70 | 35.70 |
| 10. | Pinar et al., 2017 [44] | Health personnel | Physical | X | | | 77.50 | 13.30 | 9.20 |
| | | | Verbal abuse | X | X | X | 72.60 | 18.00 | 9.40 |
| | | | Sexual harassment | X | | X | 78.50 | 21.50 | 0.00 |
| 11. | Zampieron et al., 2010 [53] | Nursing personnel | Overall Aggression | X | X | X | 79.60 | 20.40 | 0.00 |
| | | | Verbal aggression | X | X | X | 62.40 | 37.60 | 0.00 |
| 12. | Speroni et al., 2014 [54] | Nurses | Workplace Violence | X | | | 62.40 | 27.50 | 10.10 |
| 13. | Wang et al., 2022 [55] | Nurses | Workplace Violence | X | | | 62.40 | 37.60 | 0.00 |
| 14. | Weldehawaryat et al., 2020 [56] | Nurses | Workplace Violence | X | | X | 93.60 | 6.40 | 0.00 |
| 15. | Feng et al., 2022 [57] | General practitioners | Workplace Violence | X | | | 79.49 | 20.51 | 0.00 |
| 16. | Özdamar Ünal et al., 2022 [58] | Physicians and others | Workplace Violence | X | | | 61.90 | 38.10 | 0.00 |
| 17. | Turgut et al., 2021 [59] | Physicians | Workplace Violence | X | | | 77.10 | 22.90 | 0.00 |
| 18. | Vezyridis et al., 2015 [60] | Nurses and physicians | Workplace Violence | X | X | | 80.60 | 19.40 | 0.00 |
| 19. | Elston & Gabe, 2016 [61] | General practitioners | Physical assault | X | | | 82.00 | 18.00 | 0.00 |
| | | | Verbal abuse | | | | | | |
| 20. | Oguz et al., 2020 [62] | Medical and nursing personnel | Physical violence | X | | | 63.00 | 37.00 | 0.00 |
| 21. | Newman et al., 2011 [63] | Midwives, nurses & Physician | Verbal abuse | X | X | X | 22.00 | 55.00 | 23.00 |
| | | | Sexual harassment | | | | 65.00 | 20.00 | 15.00 |
| | | | Bullying | X | X | X | 55.00 | 30.00 | 15.00 |
| | | | Physical Attack | X | X | X | 55.00 | 36.00 | 9.00 |
| 22. | Arnold et al., 2020 [64] | Physicians | Sexual Harassment (W) | X | | X | 99.00 | 0.00 | 1.00 |
| | | | Sexual Harassment (M) | | | X | 49.40 | 42.50 | 8.00 |
| 23. | Benzil et al., 2020 [65] | Surgeons | Sexual harassment | | X | X | 72.00 | 5.00 | 23.00 |
| 24. | Freedman-Weiss et al., 2020 [66] | Trainee residents | Harassment (W) | | X | X | 55.40 | 44.60 | 0.00 |
| | | | Sexual Harassment (M) | | X | X | 30.30 | 69.70 | 0.00 |
| 25. | Nukala et al., 2020 [67] | Vascular trainees | Sexual harassment | | X | X | 60.00 | 40.00 | 0.00 |
| 26. | Smed et al., 2020 [68] | Vascular surgery | Sexual harassment | | X | X | 54.00 | 46.00 | 0.00 |

*(Continued)*

**Table 3.** (Continued)

| S. No | Author/s | Population | Form of Violence | Type of Perpetrators | | | Gender of Perpetrators | | |
|---|---|---|---|---|---|---|---|---|---|
| | | | | Consumers | Horizontal | Vertical | Men % | Women % | Both Genders % |
| 27. | Crebbin et al., 2015 [69] | Medical Personnel in Surgery | DBSH | | | X | 79.00 | 21.00 | 0.00 |
| | | | Sexual harassment | | | X | 90.30 | 9.70 | 0.00 |
| 28. | Jain et al., 2019 [70] | Ophthalmologists | Bullying | | | X | 79.50 | 20.50 | 0.00 |
| 29. | Lall et al., 2021 [71] | E.M. residents | Emotional abuse | X | | X | 58.20 | 41.70 | 0.00 |
| 30. | Picakciefe et al., 2017 [72] | Health personnel | Mobbing | | | X | 8.10 | 91.90 | 0.00 |
| 31. | Vorderwulbecke et al., 2015 [73] | Primary care physicians | Aggression | X | | | 80.00 | 20.00 | 0.00 |
| 32. | DiFiori et al., 2023 [172] | Surgeons, fellow and residents | Bullying | X | X | X | 67% | 33% | 0 |
| 33. | Janatolmakan et al., 2023 | Hospital nurses | Physical Violence | X | X | X | 89.2% | 10.8% | 0 |
| | | | Verbal Violence | | | | 88% | 12% | 0 |
| 34. | Zachariadou et al., 2018 [192] | Medical, & Nursing personnel | Mobbing | | | X | 19.80 | 69.10 | 11.00 |
| | **Average Perpetuation** | | | | | | **65.07** | **28.18** | **6.74** |

While the search terms yielded many studies, there was significantly less information on the gender of perpetrators of WPV. Of the 178 studies reported on perpetrators, only 34 studies provided data for perpetrators' gender (detailed in Table 3). Across the three types of violence, more men (65%) were responsible for perpetrating WPV compared to women (28%). Both men and women perpetuated violence in the remaining 7% of cases. Of the 34 studies, 25 studies reported on Type II violence, predominantly perpetrated by men, encompassing general violence [37, 40, 53–60], physical violence [6, 35, 44, 61–63, 178], verbal violence [6, 35, 44, 53, 63, 172, 178], and sexual harassment [8, 41, 44, 63, 64]. In most of these studies, women experienced a higher prevalence of violence than men. Gender-based workplace violence against nurses emerged as a pressing issue for Type II (56.2%) violence in ten studies [6, 8, 35, 40, 53–56, 168, 178]; men perpetrated 80% of the violence while women were responsible for only 19% violence, and almost all studies reported a higher prevalence of WPV against female nurses. A recent study [168] from 79 countries, though reported gender was not significant for WPV, being a nurse had higher odds of experiencing WPV (OR = 1.95; 95% CI 1.46 to 2.59, p<0.001) than a physician (OR = 1.70; 95% CI 1.33 to 2.18, p<0.001). In this study, most perpetrators were consumers (56%), followed by supervisors (16%) and colleagues (9%), or a combination of all (19%).

Violence perpetrated by colleagues (Type III-A) was reported by 15 studies, including seven for physicians [41, 43, 65–68, 172], three studies for nurses [40, 53, 178], and five that included both professionals [29, 37, 44, 60, 63]. Approximately 24% of violence was perpetrated by colleagues (Type III-A) among nurses and physicians. More perpetrators were men (63.5%) than women (23%), and some violence by colleagues was reported as perpetrated by both men and women (13.5%). Only one study [29] reported higher rates of bullying by women (37.9%) than men (10.5%) and by both genders (51.6%). Two other studies reported higher mobbing behaviours (20% Vs. 69%) (192) and (8%vs.93%) (72) by women. In these studies, most perpetrators (40.7%) were supervisors and senior colleagues (Type III-B). Victims were both physicians (53.1%) and nurses (53.6%) with similar intensity, but a higher number of women (n = 195, 56.4%) were exposed to bullying than men (n = 18, 36%). Additionally, those who experienced bullying had lower levels of psychological health status. Bullying from colleagues (26.4%) and patients/consumers (7.7%) was perceived as less harmful than

bullying from supervisors (Type III-B), which was also less reported because of the fear of consequences.

Of the 34 studies reporting on the gendered perpetuation of WPV (Table 3), 24 reported on Type III-B (vertical) violence, which was more prevalent among physicians (51.5%) than nurses (16%). When it did occur among nurses, more men (77%) perpetuated Type III-B violence than women (18%) and both men and women (5%). Several studies highlighted physicians as perpetrators of WPV against nurses regardless of gender [8, 51, 53]. Similarly, more men (67.5%) than women (24.2%) and both genders (8.2%) perpetuated Type III-B violence among physicians. In seven of ten studies (70%) for Type III-B violence among physicians, male supervisors and administrators perpetuated sexual harassment [41, 64–69]. Four studies reported bullying [43, 70, 172] and emotional abuse [71], which was also perpetrated by men.

Medical residents appear to be particularly vulnerable to Type III-B violence, with more than 60% of studies [41, 43, 64, 66, 67, 71, 172, 175] reporting this type of violence in medical residency programs. Furthermore, several studies highlighted that the perpetrator of sexual harassment was most often of the opposite sex [63, 64, 66]. For instance, Freedman-Weiss et al. [66] reported that male residents experienced 65.9% of harassment from men compared to 81.8% from women. On the other hand, female residents reported experiencing more harassment from men (97.7%) compared to women (42.4%). In the same study, the main perpetrators for female resident victims were attending physicians (72.9%), followed by nurses (68.5%), senior colleagues (44.7%) and same-level residents (23.5%). Among male residents, nurses were the most common perpetrator of WPV (69%), followed by attending physicians (62%), senior colleagues (41.9%) and same-level residents (25.6%).

Healthcare professionals in lower hierarchical positions, such as nurses and residents, often contend with stressful conditions and managerial or administrative abuse and harassment, posing challenges to patient care, institutional integrity, and the healthcare system. These experiences also detrimentally impact the victims' health and career progression. For instance, Tekin and Bulut [51] found that Turkish nurses who experienced Type III-B violence reported feelings of anger, humiliation, confusion and sadness. Moreover, these experiences also led to strained relationships with others, decreased performance, and caused them to consider leaving the profession. Although this study did not specify the gender of the offender, women experienced significantly higher verbal abuse. The highest perpetuation for all forms of abuse, including verbal (85.7%), physical (46.4%) and sexual (94.4%), was from physicians. In these cases, gender and status within the organizational hierarchy played a critical role in perpetuating Type III-A and III-B WPV, which requires serious attention from employers and health organizations to address GB-WPV through a gender-sensitive approach.

## Discussion

Our examination explores the complexities of gender dynamics concerning both the perpetrator and the victims of workplace violence within the global healthcare community, mainly focusing on nurses and physicians. While 178 studies provided information about perpetrators and sex-segregated findings for workplace violence, only 34 studies (19%) reported the gender of the perpetrator for Type II and Type III violence. These findings provided insights into how gender and an individual's position within the organization create unique vulnerabilities to WPV. The consequences of such violence against health workers not only affect patient care but also have broader implications for healthcare organizations and workforce landscapes. In our review, men were found to be the primary instigators, accounting for 65% of incidences of WPV, while women were responsible for 28% of instances. Both men and women perpetrated the remaining 7% of incidents. Additionally, our analysis identified distinctive behaviour

patterns among male and female offenders. Recognizing that each type of violence requires a different approach for its management and prevention, we will discuss the divergent behavioural patterns of men and women perpetrators of Type II and Type III violence. We examine the underlying factors contributing to these differences and discuss the implications of adopting gender-sensitive approaches to prevent and manage GB-WPV.

## Type II WPV- Client/patient

Of the 34 studies that provided the gender of perpetrators for any type/s of violence, the majority (74%, n = 25) reported on Type II WPV perpetrated by patients, their families, or visitors. In this context, male perpetrators were more prevalent, targeting both nurses (77.9%) and physicians (70%). The majority of studies that reported on Type II violence indicated a higher prevalence of various forms of violence against female nurses and physicians. The higher perpetration of WPV by men can be linked to societal norms associating aggression and dominance with masculinity [193]. At the same time, violence against a feminized nursing workforce is normalized as part of the job [24, 75, 98, 193]. This link between societal norms and assigned roles was evident in several studies [76, 125], which is deliberated in the following section.

Type II violence typically targets healthcare providers in the performance of their professional duties and is characterized by acts of physical violence [6, 35, 44, 61–63, 166]; verbal violence [6, 35, 44, 53, 63], and sexual harassment/ violence [8, 41, 44, 63, 64, 167]. Most of these studies reported a higher prevalence of WPV for women for all forms of violence [8, 9, 44, 53, 62, 64, 169, 174, 176]. The social norms, which stem from social relations dictate gender roles and responsibilities, and healthcare institutions are no exception to these forces. For example, a study conducted in Italy that included all areas of practice and the entire health workforce, investigating determinants of aggression against the health workforce reported women were 1.37 times more likely to experience aggression from consumers and colleagues. In this study, nurses experienced the highest number of episodes of violence (64%). Most of these aggressive acts occurred during assistance and supportive care to patients (38%) [125]. On the other hand, men were not immune to WPV, particularly physical [44, 61, 166, 185] and both physical and verbal violence in the emergency department in Saudi Arabia, Turkey and China [37, 59, 189]. In Turkey, male physicians experienced higher violence (62.4%) in contrast to their female counterparts (37.6%) [59]. A similar pattern emerged in Saudi Arabia, with male physicians and nurses reporting a higher prevalence (57.8%), than their female counterparts (42.8%) [37]. These three studies identified several factors for the high occurrence of WPV from patients and their relatives, including dissatisfaction with the treatment, long wait times and lack of staff [37, 59], overcrowding and lack of security [37]. Though these highlighted factors are important to explain the occurrence of workplace violence for both men and women in the workforce, in the Saudi context, culture seems to have a protective factor for women, where public abuse from men is socially unacceptable [88]. Similarly, three other studies in Jordan attributed the higher prevalence among male physicians to culture and the existence of laws that intensify legal penalties against women abusers [87], the cultural norm of altruism and tolerance towards females, particularly physical violence [42], and a lack of encouragement for reporting WPV by females as part of the male-dominant culture [150]. Additionally, the higher occurrence of physical violence for men can also be explained by the cultural expectation of masculinity.

In contrast, women's experience of severe sexual harassment was associated with pregnancy, family responsibilities, and occupational segregation [63]. Newman et al. [63] explained that occupational segregation also creates a vertical hierarchy where women are assigned to

lower-level tasks (typically front-line care providers). The WHO report analyzed gender and equity in the health and social workforce 'delivered by women, led by men' (2019) and acknowledged occupational segregation as universal, which is reinforced by the broader societal norms and creates discriminatory practices with regard to gender and occupational roles [194]. In these lower positions, women experience sexual harassment from male colleagues, male patients and community members [16, 194]. Considering the prevalence of Type II violence for both men and women linked to sex-segregated responsibilities and societal structures. Jafree [195] calls on policymakers to ensure security and protection for the health workforce, particularly women; legislative reforms for healthcare governance and zero-tolerance policies for violence were also recommended. Several other sources, too, advocate for zero-tolerance policies and emphasize the need for a managerial approach that takes all complaints seriously, reports investigation outcomes, and enforces sanctions to eliminate impunity [9, 92, 131]. Collaborative community efforts are required to acknowledge and alter the patriarchal culture and reduce violence against women by creating awareness about the public role through various forums, including the media [28, 79, 94, 195].

Several contributing factors have been identified in the context of Type II WPV, such as noise levels, inadequate communication skills [74], perceived/actual staff incompetence or unsympathetic attitudes, dissatisfaction with service provision, prolonged wait times, and poor communication [53, 196]. These circumstances can escalate emotions and increase the likelihood of violent encounters. Furthermore, specific treatment specialties, such as emergency departments [35, 75, 191], psychiatric units [76, 77], and geriatric care [26, 76], have demonstrated a higher risk of Type II workplace violence. Factors specific to these settings include a lack of privacy and personal space, unrealistic expectations of clients, insufficient staffing and resources, poor staff skills mix, healthcare systems and processes not understood by clients, perceived favouritism, overcrowding in emergency departments, delays in providing analgesia, and inflexible visiting hours [196]. These challenges, compounded by a shortage of skilled professionals, unclear expectations and communication, scheduling issues, and environmental stressors can generate increased stress and, thus, uncertainty. Addressing these factors constitutes the initial step in decreasing or eliminating the risk of violence for both men and women [197].

Both primary research and systematic reviews have acknowledged the difficulty associated with addressing multifactorial violence, given the diversity in population and setting and the types/classifications of violence [94, 95, 102, 198]. However, these sources did not provide information about perpetuators, particularly gendered nature. For instance, a recent umbrella review examined 32 systematic reviews for WPV prevalence and characteristics. This comprehensive assessment reported that the overall prevalence from the meta-analysis of 11 reviews was 57.9%, ranging from 34.1% to 78.9% among healthcare providers and most affected were nurses working in psychiatric wards [198]. This prevalence aligns with the findings of this review. Of note, the umbrella review too did not provide information on perpetrators and prevalence based on gender and stated that the included reviews had reported variable results for men and women; however, it did underscore how gender imbalances in emergency departments could increase the risk of violence among women. Several studies in our review recommended ensuring gender equality in the health workforce and leadership positions to reduce the prevalence of WPV among women [9, 30, 63, 80].

## Type III WPV-Worker-on-Worker

Type III-A (Horizontal or lateral) workplace violence perpetrated by one healthcare worker against another may stem from interpersonal conflicts, workplace stress [12], or other factors contributing to a hostile work environment. Among studies that provided data on Type III-A

violence, most perpetrators were men (63.5%) compared to women (23%). Horizontal WPV was reported more frequently by physicians [41, 43, 65–69] than among nurses [40, 53]. The studies that sampled both nurses and physicians [29, 37, 44, 60, 63] also reported that men perpetuated all forms of violence in most cases for both male and female victims [37, 44, 63]. In some instances, women experienced violence from both men and women [63].

Type III violence is also rooted in cultural norms and societal expectations that allocate roles and responsibilities based on gender [63]; in most cultures, women are responsible for childbearing and rearing and men hold decision-making positions. This phenomenon transcends the household and is also seen in the workplace and healthcare institutions [9, 68, 78]. These gendered roles and responsibilities often position men in leadership positions while women are assigned to caring roles with less authority and responsibility, perpetuating discriminatory practices that negatively impact women [9, 63, 70]. This dynamic prevails in both wealthy and lower- and middle-income countries in varied behaviors. For instance, in Australia and New Zealand, women experienced significantly higher discrimination (31% vs. 8%) and sexual harassment (23% vs. 0.5%) than men, primarily due to family responsibilities, lack of mentorship and rigid promotion criteria [70]. In Rwanda, women's experiences of childbearing and care, including managing pregnancy, motherhood and work, and the widespread negative stereotyping of women at work led to discrimination that co-occurred with sexual harassment within health workplaces [63]. Jacobson et al. [12] report on Type III-A violence in medical residency programs, and women experienced a significantly higher frequency of work-related incidents from colleagues and support staff, explaining the higher workload for women due to the coexistence of family responsibilities. Additionally, relational and managerial issues, including organizational affairs within large, complex health organizations, shifting duties and cohabitation of various teams on the same unit, were identified as factors contributing to Type III-A violence in Italy [53].

This type of interpersonal violence, including violence against women, is prevalent in science, technology, engineering and math (STEM), which are considered male-dominant disciplines [199, 200], unlike healthcare, where 70% of the workforce globally are women and higher rates of violence are associated with their roles and responsibilities and the gendered workplace hierarchy [194]. In STEM, violence against women can be explained by the backlash effect, in which gender equality is associated with higher prevalence [200].

Given the social reality of women's lives and career development in healthcare, flexible human resource development and management policies could empower women to balance their work and family responsibilities. Zampieroni et al. [53] recommend adopting realistic workloads and skill-mixed staffing, promoting gender equality in staff allocation, and participatory leadership to overcome relational conflict and managerial actions that enhance working conditions. Nurse managers must play the role of cultural gatekeepers, hold individuals accountable and foster staff empowerment; utilizing research-informed methods such as 'cognitive rehearsal and crucial conversations' [20] and conducting team-building workshops will assist in mitigating the impact of horizontal violence [21].

Type III-B (vertical) violence is primarily perpetrated by senior colleagues, supervisors, and administrative personnel occupying higher positions in the organizational hierarchy than the victim. Among the 34 studies, 66% reported perpetrators' gender for Type III-B violence; men perpetuated 77% among nurses and 67.5% among physicians. The causative factors for vertical violence included organizational structure, leadership and administrative authorities, and power struggles in the health workplaces. These factors not only perpetuated WPV but also prohibited reporting of the instances due to fear of reprisal [29, 63, 66]. Two prevalent forms of violence linked to hierarchical/ organizational structure were sexual harassment and bullying/mobbing. The majority of studies reporting Type III-B violence reported sexual violence

from male supervisors and administrators [8, 41, 44, 63–69], particularly in medical residency programs—placing these trainee residents in a vulnerable position [41, 66, 67]. Additionally, vertical violence was the only type reported to be perpetuated by women at higher levels in the organizational hierarchy, particularly bullying (women: 37.9% vs. men: 10.5%) among nurses and physicians [29]. Additionally, two studies reported higher rates of mobbing behaviours by women than men among healthcare professionals, including nurses and physicians [72, 192].

Type III-B violence is emblematic of the hierarchical and inflexible organizational culture historically dominated by male medical professionals. This stemmed from beliefs and negative stereotypes, such as women being weak, unwilling to speak up, indecisive and incompetent [63]. Additionally, perceived competence was expressed as a predictor for bullying among women [42, 153]. Such perceptions reinforce the structural power held by men, particularly with male managers and physicians. The patriarchal institutional structures provide power domination among women as well, who could use their power to oppress individuals under their control. A qualitative study [201] from Estonia exposed this dynamic of domination and sexual harassment among nurses; it highlighted the association between power and the use of sexualized language. A female nurse stated, "I am more disturbed by their patronizing behaviour"; the nurse characterizes physicians' attitude as: "I am a man, I am a physician, I can do and say whatever comes to my mind" (Nurse 18, p.30). Lamesoo [201] further explained that nurses placed themselves in the hospital hierarchy between physicians and patients and acknowledged that they could not challenge a physician's incivility. However, these nurses can easily ask patients to refrain from such behaviour without hesitation because patients have less power than nurses, and patients are expected to follow hospital rules [201] dictated by nurses. These instances explain organizational power as a protective factor for offenders. However, women's underrepresentation in positions of power places them in a vulnerable position.

Another qualitative study in Uganda by Newman et al. [9] reported from key informant interviews in the Uganda health system that "we have women over-represented in the bottom of any organization and for the men, it is an upward or inverted pyramid whereby as you go up the power ladder. . .. There is a tendency to abuse that power and they don't even think that they are abusing it because they have grown up thinking they may be flattering the women. . .". The authors further stated that "Sexual coercion started during recruitment of health workers and continued after hiring, perpetrated by men in hierarchically superior decision-making positions supervisors, senior managers (including human resources) or medical superintendents" [9]. These severe human rights violations necessitate a transformation in the mindset of individuals in the workforce and a cultural shift at organizational levels to rectify the dominant, hierarchical and permissive environment [65]. Ensuring gender equality at the upper echelons of healthcare organizations and in decision-making positions is crucial to establishing a secure and equitable environment for all, regardless of gender. A scoping review of three evidence-based guidelines and 33 systematic reviews on strategies to prevent and manage WPV in healthcare settings reported a correlation between strong leadership to cultivate and enforce a culture of inclusivity, support and respect as a prerequisite for successful prevention of WPV [202]. Therefore, healthcare organizations' leadership must proactively seek organizational solutions to end gender-based WPV and prioritize gender equality and protecting employees' rights as part of their human resources for health (HRH) policy [9].

Sexual harassment in academia was found to be an issue across various contexts, particularly among women in medical residency programs. A study [78] in a U.S. medical college reported that one-third of respondents experienced sexual harassment, including medical students (51.7%), residents/fellows (31%) and faculty members (25%), which was inversely proportional to their position in the program. Similarly, sexual harassment was more prevalent among women in vascular surgery in the U.S. [67], ophthalmology in Australia and New

Zealand [79] and cardiothoracic surgery, reported by a global survey [28], and rates of sexual harassment in almost all contexts were higher among female trainees. In one instance, in the U.S., male (70%) and female (69%) residents [41] in obstetrics and gynecology residents experienced sexual harassment at similar levels [41]. Additionally, one study in the U.S. with a large, representative sample (n = 6000) from a national survey reported that higher women's representation within a specialty was associated with lower sexual harassment for both men and women from coworkers and patients [80]. This observation held true in the Canadian context where reporting of sexual harassment incidents was low (2.9%) in a study with female participants constituting 53% of the sample [33]. These women did report slightly higher rates of intimidation, harassment, and discrimination (IHD) based on gender (males 40.4%; females 48.0%). Hence, findings underscore the recurring recommendation of gender equality in the health workforce and leadership positions and the role of leadership in preventing Type II and Type III violence, including harassment.

Acknowledging sexual harassment as a prevalent problem is the crucial initial step in formulating a successful strategy to prevent its occurrence [65, 203]; a comprehensive strategy should encompass a zero-tolerance statement across the specialty with a transparent and fair mechanism for reporting sexual harassment [65, 78]. Moreover, it is essential to provide trainees with both direct face-to-face and electronic routes for anonymous and confidential reporting to alleviate concerns related to personal reattribution and academic detriment [64, 78]. Standardized, transparent reporting mechanisms with well-delineated consequences for the offender must be established. Additionally, the institutions should ensure the availability of links to all the required resources is the first hit on online searches, displaying posters/presentations/ads [78].

Recognizing harassment as an institutional structural issue, senior leadership can have a protective role by serving as role models. A qualitative study conducted in Germany [197] representing women nurses (50%) and physicians (50%) explored preventive options for sexual harassment in academia. The findings revealed that leadership commitment and clear statements can significantly influence multiple levels by demonstrating openness to address taboo topics, raise awareness, and place the issue at the decision table. A participant stated, "A culture of political correctness is communicated from the top down, with the management committee and senior management acting as role models" (p.12). Another participant stated, "It is the senior staff that creates a team culture that should be supportive and transparent, with clear boundaries. . . .. I have an open door and open eyes policy and try to initiate rituals that allow us to work together in the correct way" (p.12). While commitment and stated actions are essential, meaningful cultural change necessitates the consistent, active, structured, and continued engagement of all health workforce members, including students and trainees, staff and especially from senior leadership. Senior leadership must be actively engaged in this process, particularly male leaders. Therefore, engaging individuals at various levels in open, nonjudgmental conversations is paramount to breaking the silence [30] and ingraining these principles into the organizational culture.

## Limitations

First, in our comprehensive review of workplace violence (WPV), not all studies reported on perpetrators of WPV. Therefore, we included all studies that indicated perpetrators/ sources of violence. We categorized these sources into distinctive categories of Type II and Type III WPV. Limitations to this approach include the heterogeneity of the forms of violence reported by the included studies according to gender. While studies reported victims' exposure to Types II, and/or III, the gender of perpetrators in each case was not specified. As a result, we

presented the prevalence of the various forms and categorized the perpetrators' type for all the studies (178) in Table 2. The final set of studies (19%) that reported on the gender of the perpetrators was analyzed. Since fewer studies provided information about the gender of perpetrators across the types/forms of violence, future research must focus on conducting and reporting gender-segregated findings for perpetrators that will strengthen recognition of the gender-based WPV and could lead to gender-sensitive strategies at the local and international levels. Another limitation of our review was that most of the included studies operationalized gender as a binary. A few studies included either non-binary (less than 4%) [80, 98] individuals or mentioned as others (less than 4%) [28, 30] or unknown (less than 9%) [85, 128], in the analysis of the total population, reported in Table 2. Even these studies did not report findings for those minority populations or address it as a limitation. Therefore, we reported findings based on gender binary. All these studies, which represented non-binary individuals, were conducted in the USA [30, 80, 98, 128] and Canada [85]; in these contexts, gender diversity and inclusion are acknowledged as compared to most Low-and -middle-income countries where sex is equated with gender. These studies did not recognize it as a limitation; only one study, which reported on survey data from the Association of American Medical Colleges (AAMC) National Sample Survey of Physicians (NSSP), expressed excluding the non-binary data because of the lower sample [80]. Considering this limitation, we recommend that future research include gender-diverse populations.

## Conclusion

The review revealed a higher prevalence of Type II and Type III WPV among women compared to men. In parallel, it was observed that men predominantly perpetrated all forms of violence against both men and women healthcare providers. Only Type III-B violence, including bullying/ mobbing, was occasionally perpetuated by women. Both Types II and Type III violence have roots in societal structures, and women were more frequently victimized. This increased victimization of women can be attributed to their lower status in society and in the healthcare settings that assign roles and responsibilities based on this status. Additionally, women's reproductive realities, including managing pregnancy, motherhood and work, and widespread negative stereotyping contributed to their vulnerability to gender-based WPV.

Conversely, men's domination in leadership, decision-making and supervisory positions in most contexts creates a hierarchical and permissive environment that perpetuates violence against women. Therefore, understanding gender implications concerning both the victim and perpetrator among the critical health workforce of nurses and physicians across the globe is essential. Healthcare organizations and professional stakeholders must seriously consider zero-tolerance policies, transparent mechanisms for handling violent incidents, and the provision of appropriate support to victims. These measures will empower individual professionals, enhance patient care, and positively impact healthcare institutions and society as a whole.

## Supporting information

**S1 Checklist. PRISMA-ScR checklist.**
(PDF)

**S1 Text. Definitions of forms of workplace violence.**
(PDF)

**S2 Text. Registered protocol.**
(PDF)

**S3 Text. Ovid MEDLINE search strategy.**
(PDF)

**S4 Text. Sources excluded.**
(PDF)

**S1 Data. Data for full text review.**
(XLSX)

## Author Contributions

**Conceptualization:** Basnama Ayaz, Andrea L. Baumann, Sioban Nelson.

**Data curation:** Basnama Ayaz, Graham Dozois.

**Formal analysis:** Basnama Ayaz, Graham Dozois, Adam Fuseini, Sioban Nelson.

**Investigation:** Graham Dozois, Andrea L. Baumann, Adam Fuseini, Sioban Nelson.

**Methodology:** Basnama Ayaz, Andrea L. Baumann, Sioban Nelson.

**Software:** Basnama Ayaz.

**Supervision:** Sioban Nelson.

**Validation:** Basnama Ayaz, Graham Dozois, Andrea L. Baumann, Sioban Nelson.

**Writing – original draft:** Basnama Ayaz, Adam Fuseini.

**Writing – review & editing:** Basnama Ayaz, Graham Dozois, Andrea L. Baumann, Sioban Nelson.

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
