## [Decision Letter · Decision Letter 0]

21 Jun 2024

PGPH-D-24-01206

Perpetrators of Gender-based Workplace Violence Amongst Nurses and Physicians – A Scoping Review of the Literature

Dear Dr. Ayaz,

Thank you for submitting your manuscript to PLOS Global Public Health. After careful consideration, we feel that it has merit but does not fully meet PLOS Global Public Health’s publication criteria as it currently stands. Therefore, we invite you to submit a revised version of the manuscript that addresses the points raised during the review process.

EDITOR: We have completed the review now. Please make sure you provide point-to-point response to all the concerns and comments from the two reviewers a a separate document.

We look forward to receiving your revised manuscript.

Kind regards,

Tanmay Bagade, Ph.D., MS (O&G), MPH, MHM

Academic Editor

Journal Requirements:

1. We note that another paper by the same author group and with the title [“A Gender-Based Review of Workplace Violence Amongst the Global Health Workforce—A Scoping Review of the Literature”; PGPH-D-23-02487] was recently accepted in PLOS Global Public Health. It has been noted that the two submissions may have some overlap with regard to the literature searches.

During your revisions, we ask that you please clarify why you feel the two reports should be considered as separate articles in your cover letter. In your manuscript, please ensure you cite (if not yet published, please upload a copy as supporting information), discuss, and acknowledge overlap with the related work, and provide adequate justification (in the Introduction) for the new submission in light of the related published work.

For more information about our policy on related manuscripts please see http://journals.plos.org/plosglobalpublichealth/s/ethical-publishing-practice#loc-submission-and-publication-of-related-studies.

2. We have noticed that you have uploaded Supporting Information files, but you have not included a list of legends. Please add a full list of legends for your Supporting Information files after the references list.

Additional Editor Comments (if provided):

Reviewers' comments:

Reviewer's Responses to Questions

**Comments to the Author**

1. Does this manuscript meet PLOS Global Public Health’s publication criteria? Is the manuscript technically sound, and do the data support the conclusions? The manuscript must describe methodologically and ethically rigorous research with conclusions that are appropriately drawn based on the data presented.

Reviewer #1: Partly

Reviewer #2: Partly

2. Has the statistical analysis been performed appropriately and rigorously?

Reviewer #1: N/A

Reviewer #2: N/A

3. Have the authors made all data underlying the findings in their manuscript fully available (please refer to the Data Availability Statement at the start of the manuscript PDF file)?

Reviewer #1: Yes

Reviewer #2: No

4. Is the manuscript presented in an intelligible fashion and written in standard English?

Reviewer #1: Yes

Reviewer #2: Yes

5. Review Comments to the Author

Reviewer #1: Workplace violence among healthcare workers is an important area of investigation. I am glad this paper attempts to synthesise findings from other studies to highlight the perpetrator's characteristics for committing Type II and Type III GB WPV among nurses and physicians. However, there are a few issues that remain unclear. I want the author to clarify those issues as follows:

1. What is GB WPV? How does one identify if a particular workplace violence is gendered or non-gendered? I think a clear conceptual definition and some ideas of indicators of what constitutes gender-based workplace violence will help the reader understand the nuances of the undercurrents of workplace violence.

2. Gender is defined in the paper in binary terms mainly because there are very few studies on workplace violence among the non-binary population. I would, however, suggest keeping the non-binary definition of gender and highlighting what those few studies have brought out and explaining why there are only a few studies on workplace violence among non-binary healthcare workers. This finding is in itself important from a gendered violence perspective.

3. The paper mainly highlights the quantitative numbers of the perpetrators of workplace violence. There is no clear analysis of who these perpetrators are besides their gender. Why did they commit workplace violence? How did the original study classify that violence- gendered or non-gendered? Did the select studies describe the nature, intensity and frequency of violence? What are the other explanatory variables besides gender that the studies included that could help identify contextual responses?

4. This paper does not dwell on the measurement of GB WPV. It is important first to do a critical analysis of how and in how many different ways GB WPV is being measured or described by different researchers. Is it 'ever' or 'last time', or do studies use a reference period -- last six months, or one year or so? How does one ensure that measurements across different studies are comparable?

5. Could GB WPV also vary according to the level of health settings? For example, nurses and physicians in primary healthcare settings could experience different kinds of pressure and expectations from clients than those working in secondary and tertiary healthcare settings. Classifying the studies according to the health settings will help us understand contextual factors that determine the GB WPV.

6. Why has this review excluded other systematic/scoping reviews? The other reviews could provide useful baseline information against which this review could compare its results.

Reviewer #2: Dear Authors,

Thank you for contributing to this growing body of work highlighting workplace related issues that often predominate, but do not receive enough attention.

This is an interesting work but does need additional attention substantiating many of the claims being made throughout, especially with regard to gender norms. I have made line by line comments in the attached version of the document. I question the helpfulness of using the workplace violence categorization given that not all of the types were used and there was a predominant focus on harassment specifically in the discussion.

Would be helpful to address how this sector relates to others with regard to workplace violence. Additionally a limitations section, gaps to address, and recommendations section is needed.

6. PLOS authors have the option to publish the peer review history of their article (what does this mean?). If published, this will include your full peer review and any attached files.

**Do you want your identity to be public for this peer review?** For information about this choice, including consent withdrawal, please see our Privacy Policy.

Reviewer #1: **Yes: **RAVI K VERMA

Reviewer #2: No

---

## [Editor Report · Decision Letter 1]

8 Aug 2024

Perpetrators of Gender-based Workplace Violence Amongst Nurses and Physicians – A Scoping Review of the Literature

PGPH-D-24-01206R1

Dear Dr. Ayaz,

We are pleased to inform you that your manuscript 'Perpetrators of Gender-based Workplace Violence Amongst Nurses and Physicians – A Scoping Review of the Literature' has been provisionally accepted for publication in PLOS Global Public Health.

Best regards,

Dr Tanmay Bagade

Academic Editor